# SEMA6A drives GnRH neuron-dependent puberty onset by tuning median eminence vascular permeability

Antonella Lettieri[1,7,10], Roberto Oleari [1,10],
Marleen Hester van den Munkhof [2,10], Eljo Yvette van Battum[2,10],
Marieke Geerte Verhagen[2,8,10], Carlotta Tacconi [3,9], Marco Spreafico [3],
Alyssa Julia Jennifer Paganoni[1], Roberta Azzarelli [4], Valentina Andre' [1],
Federica Amoruso [1], Luca Palazzolo[1], Ivano Eberini[1], Leo Dunkel [5],
Sasha Rose Howard [5,6], Alessandro Fantin [3] ✉,
Ronald Jeroen Pasterkamp [2,11] ✉ & Anna Cariboni [1,11] ✉

Innervation of the hypothalamic median eminence by Gonadotropin-Releasing Hormone (GnRH) neurons is vital to ensure puberty onset and successful reproduction. However, the molecular and cellular mechanisms underlying median eminence development and pubertal timing are incompletely understood. Here we show that Semaphorin-6A is strongly expressed by median eminence-resident oligodendrocytes positioned adjacent to GnRH neuron projections and fenestrated capillaries, and that Semaphorin-6A is required for GnRH neuron innervation and puberty onset. In vitro and in vivo experiments reveal an unexpected function for Semaphorin-6A, via its receptor Plexin-A2, in the control of median eminence vascular permeability to maintain neuroendocrine homeostasis. To support the significance of these findings in humans, we identify patients with delayed puberty carrying a novel pathogenic variant of *SEMA6A*. In all, our data reveal a role for Semaphorin-6A in regulating GnRH neuron patterning by tuning the median eminence vascular barrier and thereby controlling puberty onset.

Puberty onset and reproductive capacity depend on the projection of hypothalamic Gonadotropin-Releasing Hormone (GnRH) neurons to the median eminence (ME) and the release of GnRH into the hypothalamic-pituitary-gonadal (HPG) axis[1,2]. During development, GnRH neurons are born in the nasal placode and migrate along the terminal nerve (TN) and in association with olfactory ensheathing cells (OECs) to the medial preoptic area (MPOA) of the hypothalamus[3,4]. Here, GnRH neurons cease to migrate and extend their terminal projections to contact the ME, where complex neuro-glial-endothelial (NGE) interactions occur to regulate GnRH homeostasis[5,6].

[1]Department of Pharmacological and Biomolecular Sciences, University of Milan, Via Balzaretti 9, 20133 Milan, Italy. [2]Department of Translational Neuroscience, University Medical Center Utrecht Brain Center, Utrecht University, Universiteitsweg 100, 3584 CG Utrecht, The Netherlands. [3]Department of Biosciences, University of Milan, Via Celoria 26, 20133 Milan, Italy. [4]Wellcome - Medical Research Council Cambridge Stem Cell Institute, Jeffrey Cheah Biomedical Centre, University of Cambridge, Cambridge CB2 0AW, UK. [5]Centre for Endocrinology William Harvey Research Institute Barts and the London School of Medicine and Dentistry, Queen Mary University of London, London EC1M 6BQ, UK. [6]Department of Paediatric Endocrinology, Barts Health NHS Trust, London E1 1FR, UK. [7]Present address: Department of Health Sciences, University of Milan, Via di Rudinì 8, 20142 Milano, Italy. [8]Present address: VIB-KU Leuven, Center for Brain & Disease Research, Leuven, Belgium. [9]Present address: Division of Immunology, Transplantation, and Infectious Diseases, IRCCS San Raffaele Scientific Institute, Milan, Italy. [10]These authors contributed equally: Antonella Lettieri, Roberto Oleari, Marleen Hester van den Munkhof, Eljo Yvette van Battum, Marieke Geerte Verhagen. [13]These authors jointly supervised this work: Ronald Jeroen Pasterkamp, Anna Cariboni. ✉e-mail: alessandro.fantin@unimi.it; r.j.pasterkamp@umcutrecht.nl; anna.cariboni@unimi.it

Defective GnRH neuron migration and/or innervation of the ME leads to reduced GnRH secretion, and consequently delayed or absent puberty. These are the hallmarks of reproductive disorders such as Congenital Hypogonadotropic Hypogonadism (HH), and also contribute to the etiology of familial Delayed Puberty (DP)[7,8]. The genetic determinants underlying these reproductive disorders remain unknown in many congenital cases, suggesting the existence of mechanisms controlling the development and function of the GnRH neuronal system relevant to these conditions[9,10].

Recent studies have highlighted key roles for semaphorins and their receptors in the GnRH neuronal system[11–18]. These molecules have been implicated in GnRH neuron development and adult GnRH neuronal plasticity, which are both required to ensure GnRH secretion[19]. Consistently, pathogenic variants in various semaphorin signaling genes have been identified in patients with HH and DP[12,20–24].

Semaphorin-6A (SEMA6A) is a transmembrane semaphorin mainly known for its roles during axon guidance[25,26], neuronal positioning[27,28] and angiogenesis[29]. To exert its biological functions, SEMA6A signals as a ligand through Plexin-A2 and Plexin-A4 receptors[28,30,31]. Consistent with a role in the GnRH neuronal system, SEMA6A is expressed in the developing mouse olfactory system and hypothalamus[32,33]. However, its function in the GnRH system or in mechanisms underlying puberty onset are unknown.

Here, by combining mouse and human genetics with tailored in silico, in vitro and in vivo models, we unveil a role for SEMA6A in GnRH neuron innervation and ME vascular permeability, and identify SEMA6A as an additional player in the regulation of puberty timing.

## Results

### SEMA6A is expressed along GnRH neuron migration route during brain development

To investigate the potential role of SEMA6A in GnRH neuron development and function, we first studied SEMA6A expression in *wild-type* (WT) mouse tissues at stages relevant for GnRH neuron development and function. Double immunofluorescence staining in the nasal compartment at embryonic day (E) 12.5 and E14.5, corresponding to the start and peak of GnRH neuron migration, respectively, showed co-localization of SEMA6A and Peripherin, a TN marker (Fig. 1a, b). In contrast, GnRH neurons in the nasal compartment, identified by anti-GnRH staining, did not express SEMA6A at E12.5 and E14.5 (Fig. 1c, d). Similarly, no SEMA6A expression was found in OECs, marked with an anti-Brain lipid-binding protein (BLBP) antibody (Fig. 1e, f).

At E14.5, when the majority of GnRH neurons are scattered in the MPOA of the hypothalamus along TN fibers[14,34], SEMA6A was strongly expressed in this region but it did not co-localize with GnRH+ neurons or Peripherin+ TN fibers (Supplementary Fig. 1a, b). Next, expression of SEMA6A was examined at E18.5, when GnRH neurons have attained their final position in the hypothalamus and start to project to the ME[35]. Although GnRH+ cell bodies in the MPOA and GnRH+ neurites in the ME did not express SEMA6A (Fig. 1g, h), strong and punctate SEMA6A expression was detected in the ME in close proximity to GnRH+ axonal terminals (Fig. 1h). Our observations in mouse embryos were confirmed in human embryonic tissue at Carnegie Stage (CS) 19, corresponding to E12.5 in mouse, showing SEMA6A co-localization with axons in the nasal parenchyma (Fig. 1i–m).

Together, these data show that during GnRH neuron migration from the nasal compartment to the MPOA SEMA6A initially localizes in TN fibers, while during late gestation SEMA6A is expressed in close proximity to GnRH+ axons in the ME.

### Loss of SEMA6A does not affect GnRH neuron migration but reduces GnRH innervation of the ME

Our expression analysis suggests that SEMA6A may regulate GnRH neuron migration along nasal axons either directly or indirectly, by controlling nasal axon development, consistent with its well-characterized role as an axon guidance cue[25,26]. To explore this model, nasal axon patterning and GnRH neuron migration were analyzed in E14.5 *Sema6a+/+* and *Sema6a-/-* mouse embryos using immunohistochemistry, as described previously[14]. Unexpectedly, no obvious defects in the patterning of Peripherin+ axons were observed in *Sema6a-/-* embryos as compared to *Sema6a+/+* littermates (Fig. 2a). Also, no significant differences in the number and positioning of GnRH neurons were found (Fig. 2b, c and Supplementary Table 1).

Since the GnRH system appeared intact at E14.5, we next analyzed (1) the final positioning of GnRH neurons in the MPOA, and (2) ME innervation in E18.5 embryos, when GnRH axons can be found at or near the ME[35,36]. Although *Sema6a-/-* and *Sema6a+/+* mice displayed a similar number of GnRH neurons in the forebrain (FB) (Fig. 2d and Supplementary Table 1), GnRH neuron innervation of the ME was significantly reduced in *Sema6a-/-* mice by 2-fold (Fig. 2e, f; $p = 0.0189$; two-tailed unpaired Student's $t$ test). However, the canonical SEMA6A receptors Plexin-A2 and Plexin-A4[28,30,31] (Supplementary Fig. 2a) were not detected in MPOA-resident GnRH neurons by immunolabelling (Supplementary Fig. 2b, c). This suggests that SEMA6A does not modulate ME innervation by acting on GnRH neurons but rather through functions in other cell types.

Together these findings indicate that SEMA6A is dispensable for GnRH neuron migration and positioning, but is required for innervation of the ME by GnRH+ axons.

### Female and male mice lacking SEMA6A exhibit delayed puberty onset and sexual maturation defects

Proper innervation of the ME by GnRH axon terminals during development is crucial to ensure GnRH secretion and thereby puberty onset and reproductive function at postnatal stages[1,35]. To analyze whether SEMA6A expression in the ME is maintained postnatally, we assessed SEMA6A localization relative to GnRH axon terminals in the ME of pre- (P24) and post- (P60) pubertal mice. Interestingly, strong punctate SEMA6A expression was observed in close proximity to GnRH axon terminals in the ME at both stages (Fig. 3a, b). This suggests that SEMA6A might also play a role in maintaining the GnRH system.

Since SEMA6A was strongly expressed in the ME just before puberty (P24), we next investigated the role of SEMA6A in pubertal timing. The day of vaginal opening (VO), a proxy measurement for mouse pubertal activation of the HPG axis[35,37,38], was assessed in *Sema6a+/+*, *Sema6a+/-* and *Sema6a-/-* female mice. *Sema6a-/-* females displayed significantly delayed VO compared to WT mice, with 3 out of 9 *Sema6a-/-* females displaying a closed vaginal phenotype at the moment of termination at P45, 54 and 57 (red dots in Fig. 4a and Supplementary Fig. 3a). VO of *Sema6a-/-* mice was delayed by $12.1 \pm 2.6$ days (mean difference ± SEM) compared to *Sema6a+/+* mice, and by $5.9 \pm 0.9$ days when the 3 female mice still displaying a closed vagina at the time of sacrifice were excluded (Fig. 4a; *Sema6a+/-* $p = 0.9376$, *Sema6a-/-* $p < 0.0001$ and delayed *Sema6a-/-* only $p < 0.0001$ vs. *Sema6a+/+*; One-way ANOVA followed by Dunnett's post-hoc test). *Sema6a-/-* mice had significantly increased weight at VO as they were on average older than their WT littermates at puberty onset (Supplementary Fig. 3b). In subsequent studies, we assessed the appearance of the first estrous (FE) after VO, which corresponds to the first ovulation and indicates the reaching of sexual competence[35]. We confirmed delayed VO in this new group of mice (Fig. 4b; *Sema6a+/-* $p = 0.5186$ and *Sema6a-/-* $p = 0.0002$ vs. *Sema6a+/+*; Two-way ANOVA followed by Dunnett's post-hoc test) and detected a significant delay in FE $12.9 \pm 2.1$ days compared to *Sema6a+/+*; Fig. 4b and Supplementary Fig. 3c; *Sema6a+/-* $p = 0.5550$ and *Sema6a-/-* $p < 0.0001$ vs. *Sema6a+/+*; Two-way ANOVA followed by Dunnett's post-hoc test. One female out of 9 had not reached FE at the time of termination (P58, 24 days after VO) (Fig. 4b, red triangle).

In addition to female sexual development, male puberty onset was assessed by examining balanopreputial separation (BPS)[37,38]. Similar to

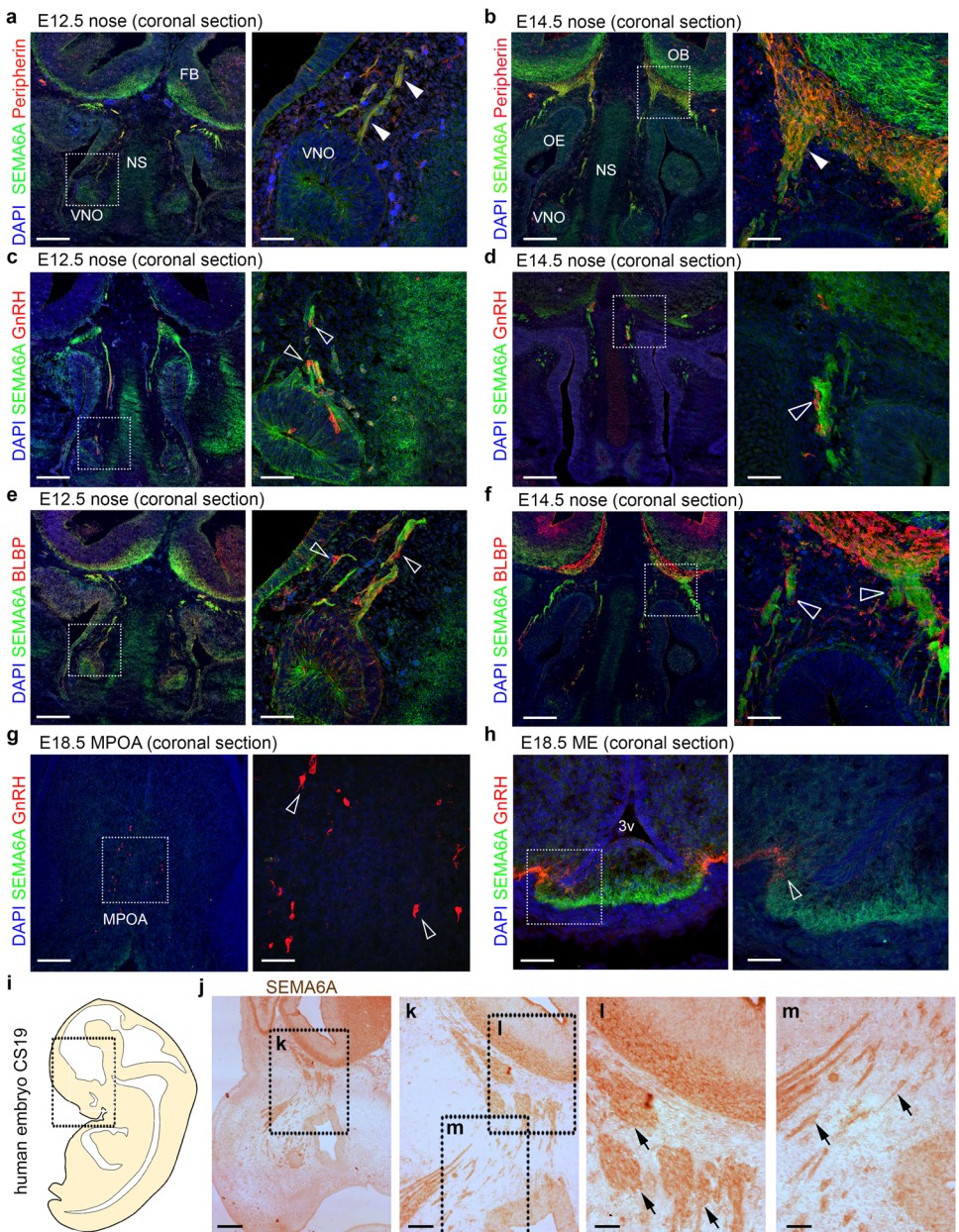

**Fig. 1 | SEMA6A is expressed in territories relevant for embryonic GnRH neuron development.** Coronal sections of E12.5 (**a**) and E14.5 (**b**) mouse heads at the nose level were immunolabelled for SEMA6A (green) and Peripherin (red). Solid arrowheads indicate examples of Peripherin-positive axons that express SEMA6A at the VNO level and in the nasal parenchyma. Coronal sections of E12.5 (**c**) and E14.5 (**d**) mouse heads at the nose level were immunolabelled for SEMA6A (green) and GnRH (red). Empty arrowheads indicate examples of GnRH neurons that lack SEMA6A expression at the VNO level and in the nasal parenchyma. Coronal sections of E12.5 (**e**) and E14.5 (**f**) mouse heads at the nose level were immunolabelled for SEMA6A (green) and BLBP (red). Empty arrowheads indicate examples of OECs that lack SEMA6A expression at the VNO level and in the nasal parenchyma. Coronal sections of E18.5 mouse brains at the MPOA (**g**) and ME (**h**) levels were immunolabelled for SEMA6A (green) and GnRH (red). Empty arrowheads indicate examples of GnRH neurons that lack SEMA6A expression in the cell body (**g**) and neurite (**h**). Sagittal view of a CS19 human embryo in a schematic drawing (**i**) and sagittal sections of schematics squared area immunolabeled for SEMA6A (**j–m**). Black arrows indicate examples of SEMA6A-expressing axons in the nasal parenchyma. Sections in (**a–h**) were counterstained with DAPI. Dotted boxes indicate areas shown at higher magnification next to the corresponding panel. Abbreviations: VNO vomeronasal organ, NS nasal septum, FB forebrain, OE olfactory epithelium, OB olfactory bulb, MPOA medial preoptic area, ME median eminence, 3v third ventricle, CS Carnegie Stage. Scale bars: 500 μm (**j**), 250 μm (**k**), 200 μm (**a, c, e, g**, low magnifications), 125 μm (**l, m**), 100 μm (**h**, low magnification), 50 μm (**a, c, e, g, h**, high magnifications).

mutant females, *Sema6a*⁻ᐟ⁻ male mice exhibited a delayed pubertal onset (3.3 ± 0.9 days compared to *Sema6a*⁺ᐟ⁺, *p* < 0.0001; Fig. 4c; *Sema6a*⁺ᐟ⁻ vs. *Sema6a*⁺ᐟ⁺ *p* = 0.0615; One-way ANOVA followed by Dunnett's post-hoc test), with 2 out of 14 males not exhibiting BPS at the day of sacrifice (P30) (Fig. 4c, red dots;). *Sema6a*⁻ᐟ⁻ and *Sema6a*⁺ᐟ⁺ males had a similar weight at BPS (Supplementary Fig. 3d).

To assess whether the reduced ME innervation observed at E18.5 persisted at later stages, we next analyzed GnRH axon terminals in the

ME of adult *Sema6a*⁻ᐟ⁻ mice. Consistent with the observed delayed puberty phenotype, all *Sema6a*⁻ᐟ⁻ females displayed a 2.6-fold decrease in ME innervation compared to age-matched *Sema6a*⁺ᐟ⁺ females (Fig. 4d, e; *p* = 0.0311; two-tailed unpaired Student's *t* test). Notably, two *Sema6a*⁻ᐟ⁻ female mice that failed to achieve VO (Fig. 4a) displayed a stronger reduction in ME innervation as compared to *Sema6a*⁻ᐟ⁻ mice exhibiting delayed VO. Similar to our observations at E18.5, the number of GnRH neurons in the MPOA was unaffected in adult *Sema6a*⁻ᐟ⁻

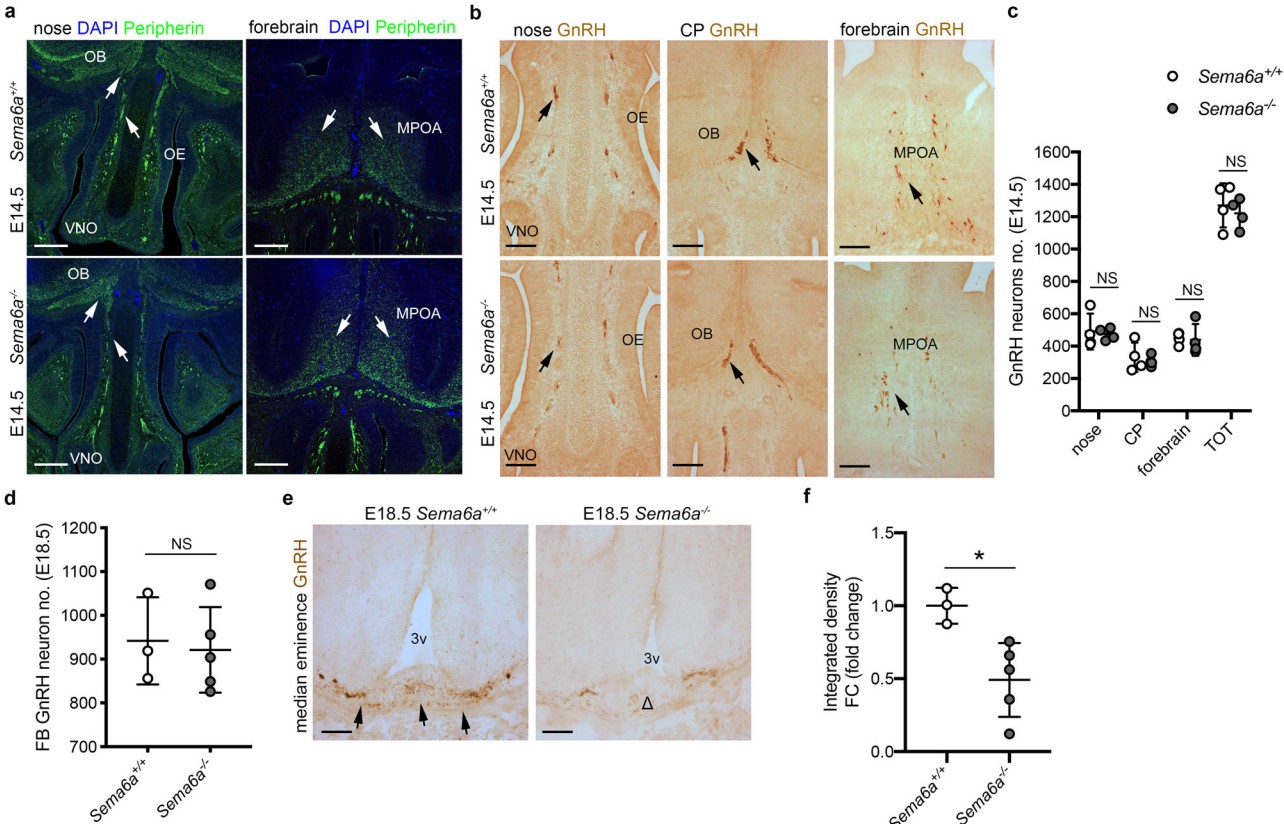

**Fig. 2 | SEMA6A loss does not impair nasal axon patterning and GnRH neuron migration, but reduces GnRH innervation of the ME. a** Coronal sections of E14.5 mouse heads of the indicated genotypes at the level of the nose and forebrain were immunolabeled for Peripherin (green) to mark nasal axons. White arrows indicate normal nasal axon development in the nose and normal TN fibers in the MPOA. Sections were counterstained with DAPI. **b** Coronal sections of E14.5 mouse heads with the indicated genotypes were immunolabelled for GnRH. Black arrows indicate examples of normal GnRH neuron distribution in the nasal parenchyma at the CP and in the forebrain. **c** Quantification of the distribution and total number of GnRH neuron in E14.5 heads ($n = 4$ per group; nose $p = 0.7974$, CP $p = 0.6360$, forebrain $p = 0.8596$, TOT $p = 0.5670$). **d** Quantification of GnRH neuron number in

E18.5 brains (from OB to MPOA) ($Sema6a^{+/+}$: $n = 3$; $Sema6a^{-/-}$: $n = 5$; $p = 0.7819$). **e** Coronal sections of E18.5 brains with the indicated genotypes at the level of the ME were immunolabelled for GnRH. Black arrows indicate normal presence of GnRH neuron axon terminals in $Sema6a^{+/+}$ embryos, whereas the decreased presence of GnRH neuron axon terminals is indicated with Δ. **f** Quantification of GnRH intensity at ME in E18.5 brains ($Sema6a^{+/+}$: $n = 3$; $Sema6a^{-/-}$: $n = 5$; $p = 0.0189$). *$p < 0.05$ and NS (not significant) after two-tailed unpaired Student's $t$ test. Abbreviations: VNO vomeronasal organ, OE olfactory epithelium, OB olfactory bulb, CP cribriform plate, FB forebrain, MPOA medial preoptic area, 3 v third ventricle. Scale bars: 200 μm (**a**), 125 μm (**b**, **e**). Data are presented as mean ± SD. Source data are provided as a Source Data file.

females (Supplementary Table 1; $p = 0.3258$). Similarly, $Sema6a^{-/-}$ males had a significant 2.2-fold decrease in ME innervation compared to age-matched $Sema6a^{+/+}$ mice (Fig. 4d, f; $p = 0.0389$; two-tailed unpaired Student's $t$ test).

We further found that adult $Sema6a^{-/-}$ females exhibited smaller ovaries but normal folliculogenesis (Fig. 4g–i; ovary weight: $p = 0.0049$; follicle subtype number/total number: primary $p = 0.7666$, secondary $p = 0.6837$, pre-antral $p = 0.3500$, early antral $p = 0.8205$; two-tailed unpaired Student's $t$). $Sema6a^{-/-}$ males displayed decreased gonadal size and a reduced number of Leydig cells, as detected by Cytochrome P450 17A1 (CYP17A1) immunostaining (Fig. 4j–l; testis weight: $p = 0.0182$; CYP17A1$^+$ area: $p = 0.0002$; two-tailed unpaired Student's $t$). Consistent with this observation, $Lhb$ mRNA levels were significantly reduced in $Sema6a^{-/-}$ pituitary glands (Fig. 4m; $p = 0.0161$). Finally, to assess general fertility, breeding records of $Sema6a$ mutant mice were compared to those of WT $C57Bl/6J$ mice for a period of 9 months (Supplementary Table 2). In this period, $Sema6a^{+/-}$ males bred with $Sema6a^{+/-}$ females produced fewer litters compared to the breeding of WT mice (68.6% successful in $Sema6a^{+/-}$ vs. 80.3% successful in WT). Crosses of $Sema6a^{-/-}$ males with either $Sema6a^{+/-}$ or WT females were only successful in 35% of the cases. Out of 9 $Sema6a^{-/-}$ males in our colony, 4 were able to produce a litter, which occurred in only 50% of their matings. Overall litter size was not affected in the

different crosses. These data support the other changes found in the reproductive system of $Sema6a$ mutants described above.

Together, these results show a requirement for SEMA6A in the innervation of the ME by GnRH neurons, and in the regulation of puberty onset and sexual maturation.

## SEMA6A is expressed by oligodendrocytes in the proximity of blood capillary loops

To establish the mechanisms through which SEMA6A controls GnRH neuron innervation and puberty onset, we first defined the cellular source of SEMA6A at the ME. Interestingly, SEMA6A signal was prominent in the basal part of the ME, which is rich in fenestrated blood capillary loops[39]. These mediate the delivery of GnRH to the pituitary, but also the entry of cues from peripheral tissues that fine tune ME homeostasis and consequently modulate GnRH structural plasticity[40–42]. Hence, we evaluated SEMA6A localization relative to blood vessels by immunolabelling SEMA6A in combination with endothelial cell (EC)[12] marker Isolectin B4 (IB4). Interestingly, SEMA6A was not detected in ECs but was concentrated and aligned to ME fenestrated blood capillary loops (Fig. 5a). To identify the source of SEMA6A in the ME, additional staining with markers for different cell types (secretory neurons, astrocytes, β-tanycytes and oligodendrocytes (OLs)) involved in ME homeostasis were performed[43,44].

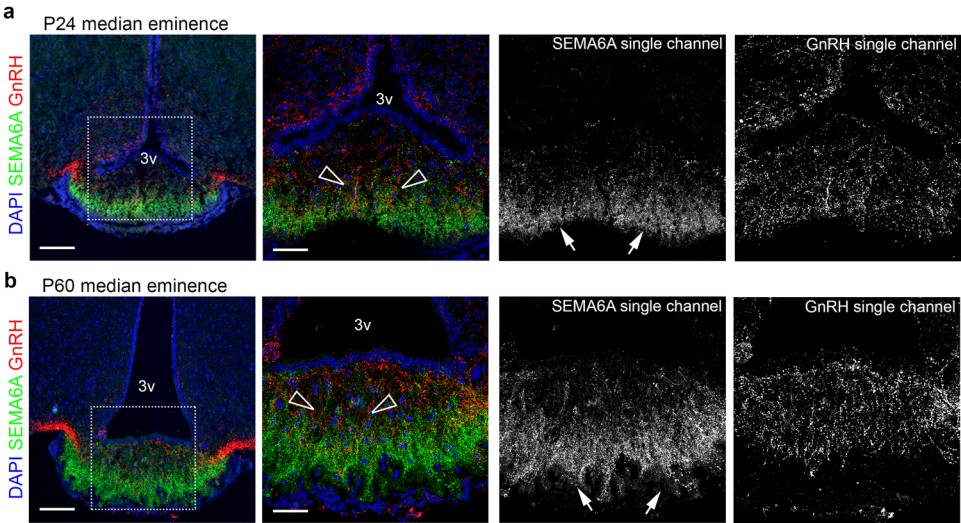

**Fig. 3 | SEMA6A is expressed in the postnatal ME in close proximity to GnRH neuron terminals.** Coronal sections of P24 (**a**) and P60 (**b**) ME were immunolabelled for SEMA6A (green) and GnRH (red) to reveal GnRH axon terminals. White boxes in (**a**, **b**) indicate areas shown at higher magnification in the adjacent panels together with single channels for SEMA6A and GnRH. Empty arrowheads indicate examples of GnRH neuron axon terminals that lack SEMA6A expression. White arrows indicate the expression of SEMA6A in the basal part of the ME. All sections were counterstained with DAPI. Abbreviations: 3v, third ventricle. Scale bars: 100 μm (low magnifications), 50 μm (high magnifications).

SEMA6A was not detected in the Synaptophysin[+] terminals of neurosecretory neurons (Fig. 5b), but surrounded these structures, as shown for GnRH-enriched terminals. Further, SEMA6A did not localize in Glial fibrillary acidic protein (GFAP)[+] astrocytes (Fig. 5c) or in Vimentin[+] β-tanycytes (Fig. 5d). Instead, and in agreement with previous reports showing that SEMA6A is required for OL differentiation[45], SEMA6A was detected in Oligodendrocyte transcription factor 2 (Oligo2)[+] cells (Fig. 5e). SEMA6A selectively localized to Glutathione S-transferase P1 (Gst P1)[+] maturing OLs but not oligodendrocyte precursor cells (OPCs) that expressed Platelet-derived growth factor receptor alpha (PDGFR-α) (Supplementary Fig. 4a, b).

Protein localization patterns were confirmed by analysis of the Tabula Muris single-cell RNA sequencing (scRNA-seq) database[46]. Dimensionality reduction combined with Uniform Manifold Approximation and Projection (UMAP) of the FACS/SMARTseq2-based dataset from adult mouse brain identified seven main clusters of cell populations: OPCs (enriched for *Olig2*, *Pdgfra*, *Cspg4* and *Sox10*), maturing OLs (enriched for *Olig2*, *Gstp1*, *Plp1*, *Cnp*, *Mog*, *Mag* and *Sox10*), OLs (enriched for *Mbp*), tanycytes/pericytes (enriched for *Vim*, *Cspg4* and *Pdgfrb*), astrocytes (enriched for *Gfap*), neurons (enriched for *Syp*, *Eno2* and *Rbpfox3*) and ECs (enriched for *Emcn*) (Supplementary Fig. 4c, e). *Sema6a* transcripts were highly enriched in premyelinating maturing OLs, with fewer transcripts in OPCs and ECs. Expression of *Sema6a* in mature OLs, tanycytes, astrocytes and neurons was very rare (Supplementary Fig. 4d).

These findings therefore suggest a possible role for SEMA6A produced by maturing OLs in the regulation of ME vascular permeability, which is crucial for proper GnRH homeostasis[5,6,39].

**Abluminal SEMA6A induces vascular permeability**

To explore the hypothesis that SEMA6A regulates ME vascular permeability, we tested the ability of SEMA6A to modulate the barrier function of EC monolayers. Trans-endothelial electrical resistance (TEER) was measured in response to SEMA6A stimulation from either the apical (luminal) or basal (parenchymal) side[47] (Fig. 6a). Although SEMA6A is produced as a transmembrane protein, its extracellular domain can be released to generate a secreted ectodomain[48]. Therefore, we transfected COS-7 cells with a vector encoding mouse SEMA6A ectodomain (SEMA6A[ECTO]) and used conditioned medium to stimulate ECs.

The ability of SEMA6A[ECTO] to affect permeability was initially examined in human umbilical vein endothelial cells (HUVECs), a well-established human model of ECs, whose sensitivity to hyperpermeability-inducing agents was confirmed by the significant TEER reduction induced by apical exposure to vascular endothelial growth factor A (VEGFA), a well-known potent inducer of vascular hyperpermeability[49–51] (Supplementary Fig. 5a). Exposure of the apical surface of the HUVEC monolayer to SEMA6A[ECTO] did not affect TEER (Supplementary Fig. 5b; SEMA6A[ECTO] apical vs mock apical, 60 min $p = 0.0011$, 120 min $p = 0.9998$, 180 min $p = 0.9810$; Two-way ANOVA followed by Tukey's post-hoc test). In contrast, TEER was significantly reduced, and hence permeability was increased, following treatment of the basal compartment with SEMA6A[ECTO], as compared to HUVECs treated with control medium (Supplementary Fig. 5b; SEMA6A[ECTO] basal vs mock basal: 60 min $p = 0.0365$, 120 min $p = 0.0006$, 180 min $p < 0.0001$; Two-way ANOVA followed by Tukey's post-hoc test). This indicates that SEMA6A can selectively induce vascular permeability in an EC monolayer.

Next, to better recapitulate ME vascular barrier in vitro, mouse brain endothelial cells (mBECs) freshly isolated from adult WT mice were analyzed[50,52]. mBECs were first validated for purity by double immunostaining for the pan-endothelial markers Platelet endothelial cell adhesion molecule (PECAM1) and Cadherin-5 (Supplementary Fig. 5c), and for responsiveness to VEGFA (Supplementary Fig. 5d). In agreement with results obtained in HUVECs, application of SEMA6A[ECTO] to the apical compartment did not affect mBEC TEER (Fig. 6b; SEMA6A[ECTO] apical vs mock apical: 30 min $p = 0.3798$, 60 min $p = 0.8985$, 120 min $p = 0.7426$, 180 min $p = 0.8563$, 240 min $p = 0.3279$; Two-way ANOVA followed by Tukey's post-hoc test). Conversely, TEER was significantly decreased when SEMA6A[ECTO] was added to the basal compartment (Fig. 6b; SEMA6A[ECTO] basal vs mock basal: 30 min $p = 0.9982$, 60 min $p = 0.0653$, 120 min $p = 0.0314$, 180 min $p = 0.0078$, 240 min $p = 0.0508$; Two-way ANOVA followed by Tukey's post-hoc test).

This indicates that parenchymal, but not luminal, SEMA6A acts as a potent inducer of endothelial permeability in vitro.

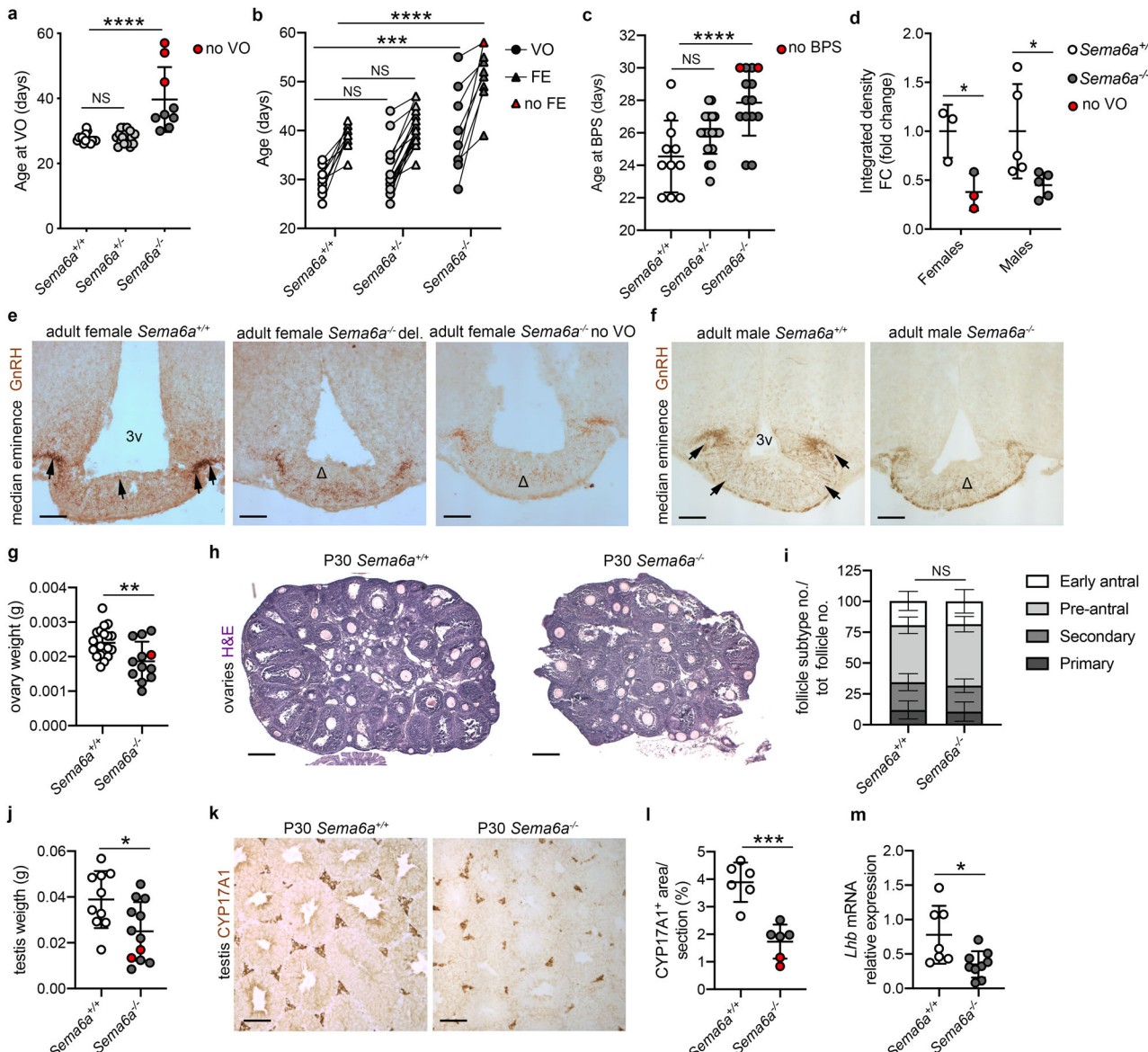

**Fig. 4 | SEMA6A loss delays puberty and gonadal maturation in female and male mice. a** Age at VO in adult females of indicated genotypes (*Sema6a*[+/+] *n* = 15; *Sema6a*[+/-] *n* = 17; *Sema6a*[-/-] *n* = 9. *Sema6a*[+/-] *p* = 0.9376, *Sema6a*[-/-] *p* < 0.0001 and delayed *Sema6a*[-/-] only *p* < 0.0001 *vs. Sema6a*[+/+]). Red dots indicate *Sema6a*[-/-] females without VO at sacrifice. **b** Ages at VO and FE in adult females of indicated genotypes (*Sema6a*[+/+] *n* = 9; *Sema6a*[+/-] *n* = 18; *Sema6a*[-/-] *n* = 9. VO: *Sema6a*[+/-] *p* = 0.5186 and *Sema6a*[-/-] *p* = 0.0002 *vs. Sema6a*[+/+]. FE: *Sema6a*[+/-] *p* = 0.5550 and *Sema6a*[-/-] *p* < 0.0001 *vs. Sema6a*[+/+]). Red triangle indicates *Sema6a*[-/-] female without FE at sacrifice. **c** Age at BPS in adult males of indicated genotypes (*Sema6a*[+/+] *n* = 11; *Sema6a*[+/-] *n* = 22; *Sema6a*[-/-] *n* = 14. *Sema6a*[+/-] *p* = 0.0615 and *Sema6a*[-/-] *p* < 0.0001 vs. *Sema6a*[+/+]). Red dots indicate *Sema6a*[-/-] males without BPS at sacrifice. **d** Quantification of ME innervation (as in **e, f**) in adult female and male brains at VO/BPS revealed a significant decreased GnRH staining in *Sema6a*[-/-] compared to *Sema6a*[+/+] in both sexes (females: *n* = 3 per group; *p* = 0.0311 males: *n* = 5 per group; *p* = 0.0389). Red dots indicate *Sema6a*[-/-] females without VO. Coronal sections of female (**e**) and male (**f**) ME immunolabelled for GnRH. Presence of GnRH neuron axon terminals in *Sema6a*[+/+] brains (black arrows) is reduced in *Sema6a*[-/-] (Δ).

**g–i** Weight (**g**) and H&E histological analysis (**h**) of adult female ovaries. *Sema6a*[-/-] mice exhibited significantly smaller ovaries (*Sema6a*[+/+] *n* = 20, *Sema6a*[-/-] *n* = 12; *p* = 0.0049) but normal folliculogenesis (**i**, *n* = 6 per group; primary *p* = 0.7666, secondary *p* = 0.6837, pre-antral *p* = 0.3500, early antral *p* = 0.8205). Red dot indicates *Sema6a*[-/-] female without FE. **j–l** Weight (**j**) and immunostaining for Leydig cell marker CYP17A1 (**k**) of adult male testes. *Sema6a*[-/-] mice exhibited significant smaller testes (*Sema6a*[+/+] *n* = 10, *Sema6a*[-/-] *n* = 12) and reduced number of Leydig cell evaluated as CYP17A1 area (**l**, *n* = 6 per group; *p* = 0.0002). Red dots indicate *Sema6a*[-/-] males without BPS at sacrifice. **m** RT-qPCR analysis for *Lhb* transcript in male pituitaries (*Sema6a*[+/+] *n* = 7, *Sema6a*[-/-] *n* = 9; *p* = 0.0161) calculated relative to controls using *Gapdh*-normalized Cq threshold values. **p* < 0.05, ***p* < 0.01, ****p* < 0.001, *****p* < 0.0001 and NS (not significant) after One- (**a, c**) or Two-way (**b**) ANOVA followed by Dunnett's post-hoc test, or Two-tailed unpaired Student's *t* test (**d, g, i, j, l, m**) Abbreviations: VO vagina opening, FE first estrous, BPS balanopreputial separation, 3v third ventricle, del. delayed. Scale bars: 500 μm (**h**), 250 μm (**e, f, k**). Data are presented as mean ± SD. Source data are provided as a Source Data file.

## SEMA6A regulates vascular permeability via Plexin-A2

As a first step to determine the signaling pathway by which SEMA6A controls EC permeability, the expression of putative SEMA6A receptors in ECs was examined in two publicly available scRNA-seq databases. As Plexin-A2 and Plexin-A4 are canonical SEMA6A receptors[28,30,31] (Supplementary Fig. 2a), our analysis focused on class A

plexins. For analysis of the Tabula Muris dataset, the FACS/SMART-seq2-based dataset was used and the cluster containing *Emcn*[+] ECs was selected (Supplementary Fig. 4c). *Plxna2*, but not *Plxna1, Plxna3* or *Plxna4*, mRNA was present at high levels and frequency (Supplementary Fig. 5e, f). Analysis of the EC atlas[53], obtained by droplet-based sequencing of FACS-isolated PECAM1[+] ECs, allowed separation of two

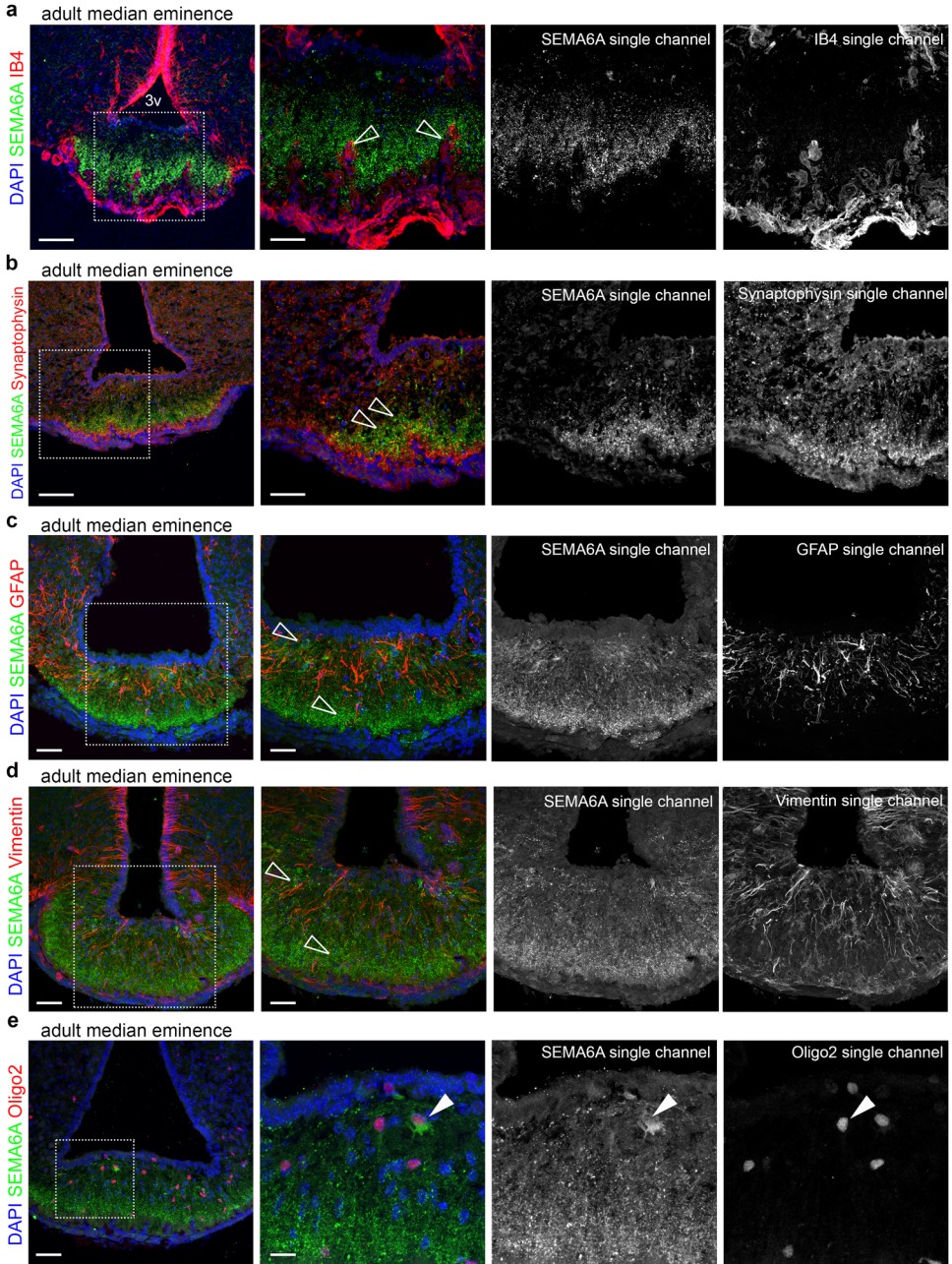

**Fig. 5 | SEMA6A is detected in OLs in the ME and not by other cell types composing the NGE unit. a–e** Coronal sections of adult ME were immunolabelled for SEMA6A (green) together with IB4, Synaptophysin, GFAP, Vimentin and Oligo2 (red) to label blood vessels, neuroendocrine neurons, astrocytes, tanycytes and oligodendrocytes, respectively. White boxes indicate areas shown at higher magnification next to each corresponding panel together with single channels for SEMA6A and different cell markers. Empty arrowheads indicate examples of blood vessels penetrating in the ME that lack SEMA6A expression (**a**) and the lack of SEMA6A expression on neuroendocrine axon terminals (**b**), astrocytes (**c**), tanycytes (**d**). Solid arrowheads indicate the expression of SEMA6A on oligodendrocytes (**e**). All sections were counterstained with DAPI. Abbreviations: 3v third ventricle. Scale bars: 100 μm (**a**, **b** low magnifications), 50 μm (**a**, **b**, high magnifications; **c**–**e**, low magnifications), 25 μm (**c**, **d**, high magnifications), 12 μm (**e**, high magnification).

discrete EC clusters thanks to the higher number of EC transcriptomes obtained despite a lower sequencing depth compared to Tabula Muris. One cluster was composed of arterial, capillary and venous ECs, whereas a smaller subset comprised *Plvap*⁺ ECs, likely representing fenestrated capillary ECs (fECs) such as those in the ME (Fig. 6c). *Plxna2* was detected in the different EC subtypes and was enriched in fECs (Fig. 6c, d). In contrast, *Plxna1*, *Plxna3* and *Plxna4* transcripts were not detected in any EC subpopulation (Fig. 6e). Analysis of the BulkECexplorer[54] confirmed that *Plxna2* transcript levels were on average ~10-fold higher than those of *Plxna1*, *Plxna3* or *Plxna4* in both HUVECs and mBECs (Supplementary Fig. 5g). These results were validated by RT-qPCR in HUVECs (Supplementary Fig. 5h). Together, these observations identify Plexin-A2 as the likely binding partner of SEMA6A on ECs.

In line with the observation that SEMA6A induced vascular permeability only when exposed to EC monolayers from the abluminal side (Fig. 6b), Plexin-A2 did not colocalize with the luminal/apical membrane marker Intercellular adhesion molecule 2 (ICAM-2)[55], but rather localized to the basal side of the monolayer (Fig. 6f, g). Finally, to establish a functional requirement for Plexin-A2 in SEMA6A-induced vascular permeability, *PLXNA2* expression was knocked down in HUVECs using *sh*RNA approaches (Supplementary Fig. 5i).

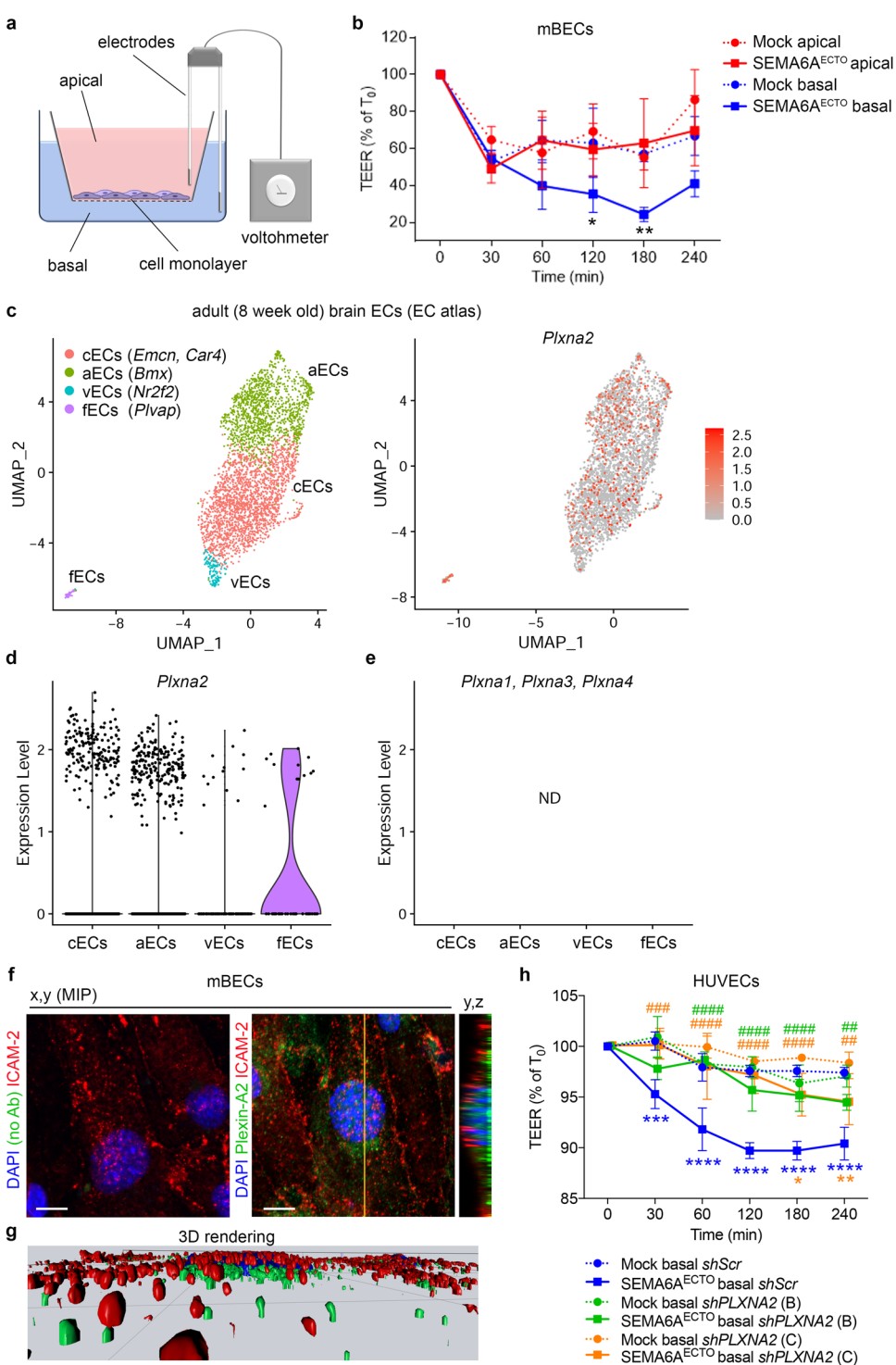

As expected, basal SEMA6A$^{ECTO}$ exposure significantly and steadily reduced TEER compared to mock treatment in HUVEC monolayers infected with scrambled control short hairpins (*shScr*) (Fig. 6h; *shScr*, blue symbols: SEMA6A$^{ECTO}$ basal vs. mock basal, 30 min $p = 0.0001$, 60–240 min $p < 0.0001$; Two-way ANOVA followed by Tukey's post-hoc test). Instead, treatment with SEMA6A$^{ECTO}$ from the basal compartment in HUVECs with decreased levels of Plexin-A2 no longer affected TEER (Fig. 6h; *shPLXNA2* (B), green symbols: SEMA6A$^{ECTO}$ basal vs. mock basal, 30 min $p = 0.0517$, 60 min $p = 0.9994$, 120 min $p = 0.3132$, 180 min $p = 0.8817$, 240 min $p = 0.1738$; Two-way ANOVA followed by Tukey's post-hoc test) or caused only a slight and delayed TEER reduction (Fig. 6h; *shPLXNA2* (C), orange symbols: SEMA6A$^{ECTO}$

basal vs. mock basal, 30 min $p > 0.9999$, 60 min $p = 0.5015$, 120 min $p = 0.7981$, 180 min $p = 0.0149$, 240 min $p = 0.0089$; Two-way ANOVA followed by Tukey's post-hoc test). Still, SEMA6A$^{ECTO}$ efficacy in reducing TEER was significantly reduced in both *shPLXNA2*-infected HUVECs compared to *shScr* HUVECs at all time points (Fig. 6h, hashtags; SEMA6A$^{ECTO}$ *shScr* vs SEMA6A$^{ECTO}$ *shPLXNA2* (B): 30 min $p = 0.2205$, 60–180 min $p < 0.0001$, 240 min $p = 0.0051$; SEMA6A$^{ECTO}$ *shScr* vs SEMA6A$^{ECTO}$ *shPLXNA2* (C): 30 min $p = 0.0006$, 60–180 min $p < 0.0001$, 240 min $p = 0.0049$; Two-way ANOVA followed by Tukey's post-hoc test). Together, these experiments show that SEMA6A can regulate barrier opening in EC monolayers via Plexin-A2.

**Fig. 6 | Abluminal SEMA6A regulates vascular permeability via Plexin-A2.** **a** Schematic of vascular permeability measurement by TEER. Apical (red) and basal (blue) compartments represent luminal and parenchymal sides of a blood vessel, respectively. Created with BioRender.com. **b** TEER quantification in mBECs apically (red) or basally (blue) treated with conditioned media from SEMA6A[ECTO] or mock-transfected COS-7 cells. TEER values are expressed as percentage of TEER at $T_0$, ranging 46–80 $\Omega cm^2$. Graph shows one out of $n = 2$ independent experiments ($n = 3$ per group; SEMA6A[ECTO] basal vs mock basal: 30 min $p = 0.9982$, 60 min $p = 0.0653$, 120 min $p = 0.0314$, 180 min $p = 0.0078$, 240 min $p = 0.0508$). **c–e** scRNA-seq analysis of adult mBECs from the EC atlas dataset. UMAP plots show distinct cell types and *Plxna2* transcript levels (**c**). Violin plots compare *Plxna2* transcript levels in different EC subsets (**d**). *Plxna1*, *Plxna3* and *Plxna4* transcripts could not be detected (ND) in any EC subpopulation (**e**). **f, g** X,y maximum intensity projections (MIP) of mBECs immunolabelled for Plexin-A2 (green) and luminal EC marker ICAM-2 (red) counterstained with DAPI, including control staining for the secondary antibody used to detect Plexin-A2 (anti-goat). Yellow line indicates the single optical y,z cross section displayed on the side (**f**). 3D surface rendering of the z-stack used to generate the x,y MIP is shown in (**g**). **h** TEER quantification in HUVECs treated basally with conditioned media from SEMA6A[ECTO] or mock-transfected COS-7 cells, following knockdown of *PLXNA2* (shRNA B, green; C, orange) or in controls (shScr, blue). TEER values are expressed as percentage of TEER at $T_0$, ranging 36–44 $\Omega cm^2$. Graph shows one out of $n = 2$ independent experiments ($n = 3$ per group). * refers to SEMA6A[ECTO] vs mock (shScr: 30 min $p = 0.0001$, 60–240 min $p < 0.0001$. shPLXNA2 (B): 30 min $p = 0.0517$, 60 min $p = 0.9994$, 120 min $p = 0.3132$, 180 min $p = 0.8817$, 240 min $p = 0.1738$; shPLXNA2 (C): 30 min $p > 0.9999$, 60 min $p = 0.5015$, 120 min $p = 0.7981$, 180 min $p = 0.0149$, 240 min $p = 0.0089$.). # refers to SEMA6A[ECTO] shScr vs SEMA6A[ECTO] shPLXNA2 (shPLXNA2 (B): 30 min $p = 0.2205$, 60–180 min $p < 0.0001$, 240 min $p = 0.0051$; shPLXNA2 (C): 30 min $p = 0.0006$, 60–180 min $p < 0.0001$, 240 min $p = 0.0049$.) *$p < 0.05$, ** or ##$p < 0.01$, *** or ###$p < 0.001$ and **** or ####$p < 0.0001$ after Two-way ANOVA followed by Tukey post-hoc test. Scale bars: 8 µm (**f**). Abbreviations: mBECs mouse brain endothelial cells, aECs arterial ECs, cECs capillary ECs, vECs venous ECs, fECs fenestrated capillary ECs, UMAP uniform manifold approximation and projection, ND not detected. Data are presented as mean ± SD. Source data are provided as a Source Data file.

## SEMA6A regulates ME capillary fenestration and vascular permeability in vivo

Having found that parenchymal SEMA6A was able to bind PLXNA2 on the abluminal side of EC monolayers in vitro to promote endothelial permeability suggests that in vivo SEMA6A, which is expressed in close proximity of blood capillary loops (Fig. 5a), may play a role in regulating the ME vascular barrier. In agreement with this hypothesis, PECAM1[+] capillary loops in the ME expressed Plexin-A2 (Fig. 7a). Further, transcardial perfusion of Evans Blue (EB) dye, to mimic peripheral signals diffusing through the ME barrier in vivo, led to EB extravasation in the ME of control mice. In contrast, EB leakage was significantly less in *Sema6a*[-/-] MEs (Fig. 7b, c; $p = 0.0212$; two-tailed unpaired Student's *t* test). To assess whether this reduction in ME barrier permeability was associated with structural changes, the number of Plasmalemma Vesicle Associated Protein (PLVAP)[+] fenestrated capillary loops was quantified[56–58]. Notably, *Sema6a*[-/-] mice had significantly fewer PLVAP[+] loops invading the ME parenchyma, as compared to WT mice (Fig. 7d, e; $p = 0.023$; two-tailed unpaired Student's *t* test). Double staining for PLVAP and PECAM1 showed reduced PLVAP immunostaining and a decrease in PECAM1[+] capillary loops in the ME of *Sema6a*[-/-] mice (Supplementary Fig. 6a). This indicates that reduced ME barrier permeability caused by SEMA6A loss is accompanied by defective structural remodeling of ME capillaries. The morphology of Vimentin[+] β-tanycytes, known to modulate NGE unit plasticity and consequently GnRH secretion[17], was unchanged in *Sema6a*[-/-] mice (Supplementary Fig. 6b).

Together, these findings show that SEMA6A is required in vivo to maintain ME barrier permeability and the formation of fenestrated capillary loops in the ME.

## Exome sequencing of patients with DP identifies a loss-of-function SEMA6A variant in six individuals

In view of the observation that *Sema6a*[-/-] mice present with delayed pubertal timing, we interrogated exome sequencing data from a large cohort of patients with self-limited DP and identified a potentially pathogenic, variant in *SEMA6A* (ENSG00000092421, NCBI gene ID: 57556). This variant (NM_020796; c.1268 T > C; p.I423T) was carried by six affected individuals from our cohort, including four members of one family, and all individuals were heterozygous for this variant (Fig. 8a). The presence of this *SEMA6A* variant in probands and family members was confirmed by Sanger sequencing (Supplementary Fig. 7). The affected subjects did not carry any other predicted pathogenic variants in known HH/DP causing genes nor other pituitary hormone deficiencies were excluded.

All the individuals (both males and females) carrying the *SEMA6A* variant showed classical clinical and biochemical features of self-

limited DP, with delayed onset of Tanner stage 2 and delayed peak height velocity. Proband A is a male who presented at 14.5–15.0 years with testicular volumes of 3.7 mL, bone age delayed by 3.6 years and short stature (height SD score −2.75) (see Table 1 for detailed auxological data). He had normal olfaction and no evidence of chronic illness. His short stature was related to pubertal delay, as height SDS at both 6.5–7.0 years and at adult height was −1.3, in keeping with his midparental target height. His delayed puberty was inherited from his mother, who had menarche aged 15.0–16.0 years; his brother and maternal uncle also had delayed onset of puberty and growth spurt. Both the proband and his brother had spontaneous pubertal development without testosterone therapy on follow up, thus excluding HH. The second male proband (B) presented at 14.5–15.0 years with delayed pubertal onset (testicular volumes 3.5 mL and lack of secondary sexual characteristics). He had a family history of delayed puberty in his mother and maternal grandfather. Proband C, also male, was first seen at age 15.0–15.5 years with testicular volumes of 4.5 mL and bone age delay of 2.8 years. Both probands B and C had progressed through spontaneous puberty by the age of 18 years, confirming a self-limited DP phenotype. DNA was not available from family members of these two probands.

## The SEMA6A[I423T] variant is predicted to affect protein stability

The identified *SEMA6A* variant was not found in the gnomAD browser (accessed 28.06.2023). The substitution of the non-polar amino acid isoleucine (Ile, I) with a neutral polar amino acid, threonine (Thr, T), is predicted to be damaging to protein function or disease-causing by 3/3 bioinformatic tools (Poly-Phen, SIFT and REVEL) with a combined annotation dependent depletion (CADD) score of 31 (Fig. 8b). The affected I423 residue lies in the SEMA domain and is highly conserved between species (Fig. 8c, d).

To further predict the possible damaging impact of the I423T variant on protein stability, an in silico analysis was performed starting from a CryoEM-resolved structure of SEMA6A in complex with the *C. sordellii* lethal toxin, TcsL (PDB ID: 6WTS)[59]. Within the SEMA domain, I423 lies close to the central cavity of the beta-sheet structure of the 6th beta-propeller blade (Fig. 8e, f), which has no direct contact to the SEMA6A dimerization or SEMA6A::Plexin-A2 interaction surfaces[59]. Since I423 forms hydrophobic interactions with some of the surrounding residues (e.g., A298, T300, L312, A313, T314), an Ile to Thr substitution would reduce the size of the side chain and its hydrophobicity, thereby reducing the extent of these interactions hydrophobicity from 4.5 for Ile to −0.7 for Thr, according to previous reports[60] (Fig. 8g). Additionally, substitution with a smaller residue would make this local niche part of the large solvent accessible cavity of the SEMA domain, leading to a possible

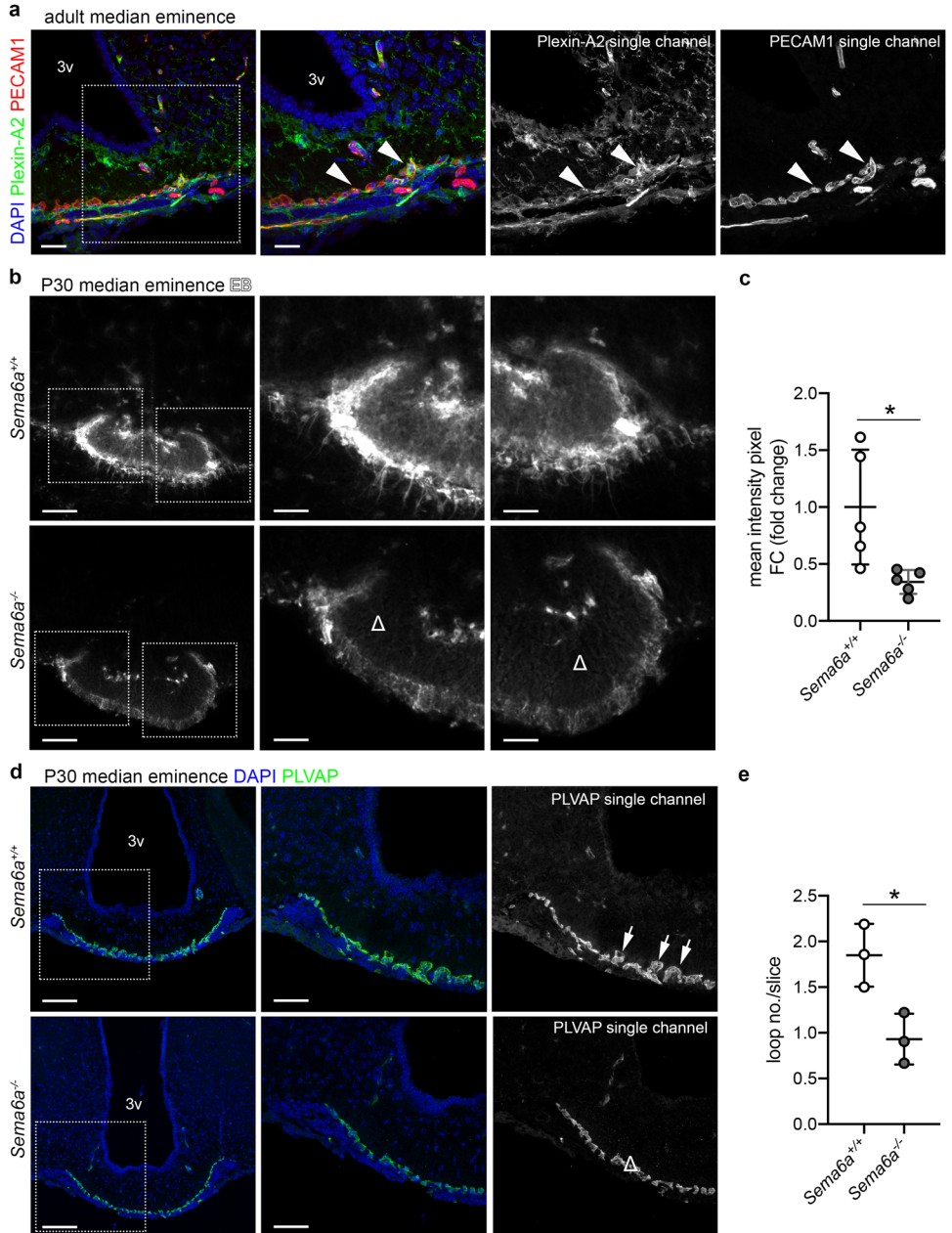

**Fig. 7 | SEMA6A loss prevents ME barrier opening and induces structural rearrangements in the ME capillary bed. a** Coronal sections of adult ME were immunolabelled for Plexin-A2 (green) together with PECAM1 (red) to label blood vessels. The white box indicates the area shown at higher magnification next to each corresponding panel together with single channels for Plexin-A2 and PECAM1. Solid arrowheads indicate the expression of Plexin-A2 on capillaries of the basal portion of the ME. **b** Coronal sections of brains from EB-injected females of the indicated genotypes. White boxes indicate areas shown at higher magnification next to each corresponding panel. Δ indicates areas of no EB leakage in *Sema6a⁻/⁻* ME parenchyma. **c** Quantification of EB diffusion in the ME parenchyma of female brains ($n = 5$ per group; $p = 0.0212$). **d** Coronal sections of female brains with the indicated genotypes were immunolabelled for PLVAP (green) to reveal capillary loops invading the ME parenchyma. White boxes indicate areas shown at higher magnification next to each corresponding panel together with a single channel for PLVAP. White arrows or Δ indicate the normal or decreased presence of capillary loops in *Sema6a⁺/⁺* and *Sema6a⁻/⁻* ME parenchyma, respectively. **e** Quantification of capillary loops in the ME of female brains ($n = 3$ per group; $p = 0.0230$). All sections were counterstained with DAPI. *$p < 0.05$ after two-tailed unpaired Student's *t* test. Abbreviations: 3v third ventricle, EB Evans Blue. Scale bars: 100 μm (**b**, **d** low magnifications), 50 μm (**a**, low magnification; **b** and **d**, high magnifications), 25 μm (**a**, high magnification). Data are presented as mean ± SD. Source data are provided as a Source Data file.

decrease in protein stability. Therefore, the impact of Ile to Thr substitution on protein stability of both SEMA6A chains was evaluated in silico with the BioLuminate *Residue Scan* tool. This analysis showed that the stability of the mutated SEMA6A homodimer was substantially decreased compared to the WT SEMA6A homodimer (ΔStability: 16.22 ± 0.14 kcal/mol for each single SEMA6A protomer), supporting an effect of the identified variant on SEMA6A stability.

## The SEMA6A^I423T variant affects protein levels and localization in vitro

To functionally validate the in silico modeling predictions, we generated myc-tagged vectors encoding WT and mutated forms of human *SEMA6A* (*SEMA6A^WT* and *SEMA6A^I423T* respectively) followed by expression in COS-7 cells. Then, protein synthesis and cellular localization were analyzed as previously described for other semaphorins at 48 h post-transfection[12,18]. Western blotting analysis revealed that both

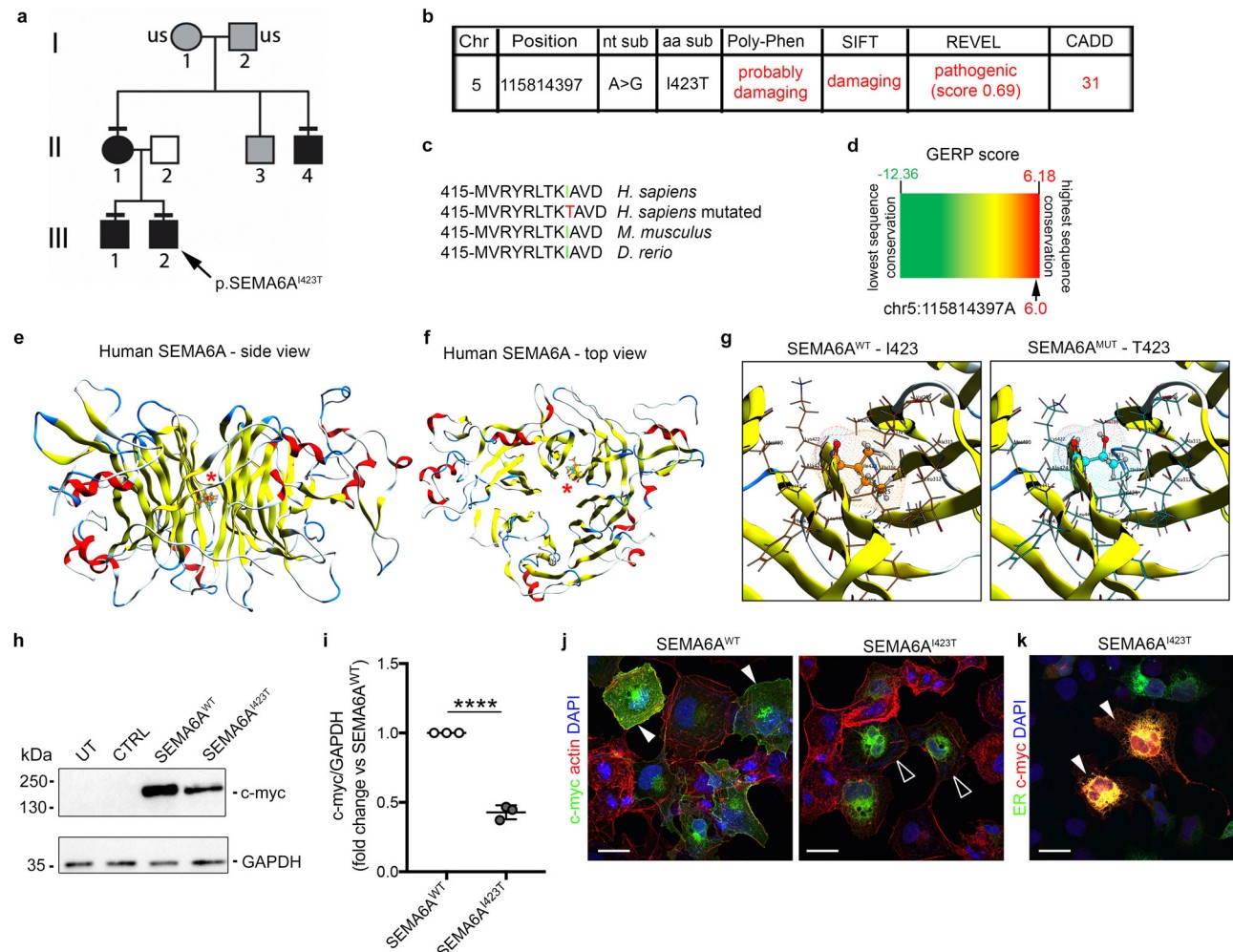

**Fig. 8 | Exome sequencing identifies a SEMA6A variant with effects on protein stability in DP patients. a** Family tree of proband A (III.2, arrow) presenting DP. Symbols: DP phenotype (black), unknown phenotype (gray); unaffected individual (white). Horizontal black lines indicate the presence of heterozygous p.I423T variant identified by WES and verified by Sanger sequencing. US signifies that DNA was unavailable for sequencing. **b** Chromosome (Chr) position, nucleotide substitution (nt sub), amino acid substitution (aa sub), and bioinformatic predictions of the p.I423T *SEMA6A* variant according to Poly-Phen, SIFT, REVEL and CADD software. **c** Partial protein sequence alignment of vertebrate SEMA6A orthologs shows that I423 conservation in mouse and zebrafish. **d** Genomic evolutionary rate profiling of sequence constraint for SEMA6A variant using GERP + + analysis (RS score = 6.0). Side (**e**) and top (**f**) 3D views of modeled human SEMA6A monomer. Red asterisks indicate the I423 residue within the central cavity of SEMA domain 6th beta-propeller blade. **g** High magnification images of I423 and T423 residues and their interactions. **h** Lysates from untransfected (UT) COS-7 cells and cells transfected with an empty control expression vector (CTRL) or vectors encoding c-myc tagged human SEMA6A[WT] or mutant SEMA6A[I423T] were immunoblotted for c-myc to recognize human (SEMA6A: 120 kDa). GAPDH (37 kDa) was used as a loading control. **i** Quantification of relative abundance of SEMA6A[I423T] protein compared to SEMA6A[WT] after normalization with GAPDH ($n = 3$ independent experiments; $p < 0.0001$). **j** COS-7 cells transfected with vectors encoding c-myc tagged SEMA6A[WT] or SEMA6A[I423T] were immunolabeled for c-myc (green) and F-actin (red). Solid arrowheads indicate cell surface SEMA6A[WT] expression, whereas empty arrowheads indicate lack of surface SEMA6A[I423T] but intracellular and perinuclear expression. **k** COS-7 cells co-transfected with vectors encoding SEMA6A[I423T] and ER-emerald, visualizing ER (green), were immunolabeled for c-myc (red) and counterstained with DAPI. Solid arrowheads indicate examples of SEMA6A[I423T] ER retention. ****$p < 0.0001$ after two-tailed unpaired Student's $t$ test. Abbreviations: ER endoplasmic reticulum. Scale bars: 50 μm (**j**, **k**). Data are presented as mean ± SD. Source data are provided as a Source Data file.

---

SEMA6A[WT] and SEMA6A[I423T] proteins were successfully synthetized, although expression of the mutant protein was significantly reduced as compared to SEMA6A[WT] (Fig. 8h, i; $p < 0.0001$; two-tailed unpaired Student's $t$ test). This suggests that the mutation affects protein expression and/or stability.

To examine whether the variant could also affect protein localization, we performed immunofluorescence using an anti-c-myc antibody to visualize SEMA6A[WT] or SEMA6A[I423T] in transfected cells (Fig. 8j). SEMA6A[WT] protein was detected at the cell membrane, as previously reported[61], while SEMA6A[I423T] protein mainly localized in the perinuclear region. To understand the subcellular localization of mutant SEMA6A, COS-7 cells were co-transfected with SEMA6A[I423T] and a fluorescent endoplasmic reticulum (ER)-marker (mEmerald-ER-3).

High level of signal colocalization suggests that SEMA6A[I423T] was retained in the ER (Fig. 8k). Together, the in vitro results strongly support our in silico predictions and indicate that the I423T variant might affect SEMA6A stability and impair intracellular protein trafficking. The consequently defective membrane localization would hamper any paracellular interaction with SEMA6A receptors, suggesting a loss-of-function effect of the SEMA6A[I423T] variant in patients.

## Discussion

Our study provides insights into the mechanisms that regulate puberty onset by implicating SEMA6A in the control of ME homeostasis and GnRH neuron plasticity and by identifying human patients with delayed onset of puberty carrying a pathogenic *SEMA6A* variant.

**Table 1 | Clinical data of auxological measurements and biochemical investigations from mid-childhood to adult age for the proband from the index family (proband A)**

| | Proband A | | | | |
|---|---|---|---|---|---|
| Age (years) | 6.5–7.0 | 14.5–15.0 | 15.0–15.5 | 16.0–16.5 | 18.0–18.5 |
| Height (cm) | 115.0 | 144.5 | 148.5 | 154.5 | 168.0 |
| Height SD | −1.3 | −2.7 | −2.8 | −2.5 | −1.3 |
| BMI | - | 23.0 | - | 23.7 | - |
| Basal LH (IU/L) (0.1–0.6) | - | 0.3 | - | 4.7 | - |
| Basal FSH (IU/L) (0.1–0.9) | - | 0.8 | - | 3.6 | - |
| Testosterone (nmol/L) | - | 0.2 | 2.1 | 10.2 | - |
| Peak Height velocity (cm/year) | - | - | - | 10.2 | - |
| Age at peak height velocity (years) | - | - | - | 16.0 | - |

Age is given as range rather than precise decimal years in accordance with journal policy.

Similar to other members of the semaphorin family, *SEMA6A* regulates axon guidance during brain development[25,62]. In line with this, we detected strong SEMA6A expression in nasal axons of developing mouse and human embryos (Fig. 1). However, *SEMA6A* loss did not affect GnRH axon pathfinding or GnRH neuron migration (Fig. 2). These results indicate a dispensable role for *SEMA6A* in these processes, as already observed for other semaphorins (e.g., *SEMA3F*)[11]. In addition to their key roles in axon guidance and neuronal migration at embryonic stages, semaphorins are expressed in the postnatal hypothalamus where they control GnRH neuron structural plasticity[15–17]. We found that *SEMA6A* is stably expressed in the ME parenchyma of perinatal (E18.5) and postnatal mice (P24 and P60) (Figs. 1, 3), a region that undergoes several cellular rearrangements to regulate neurohormonal homeostasis[39]. Intriguingly, neither *SEMA6A* transcript nor protein were detected in neuroendocrine neurons, β-tanycytes or astrocytes (Fig. 5 and Supplementary Fig. 4). These cells are known to play key roles in GnRH neuron plasticity[43,44]. Instead, we identified maturing OLs as the source of *SEMA6A* in the ME (Fig. 5 and Supplementary Fig. 4). Thus, in addition to its previously described role in OL differentiation[45], *SEMA6A* derived from OLs in the ME serves as a signal for regulating ME vascular permeability and GnRH neuron plasticity.

*Sema6a*[-/-] mice displayed decreased GnRH axon innervation of the ME starting from E18.5 (Fig. 2), when the first GnRH[+] fibers project to the ME[35], until adult stages (P60) (Fig. 4), when GnRH neuron plasticity must be tightly regulated to guarantee proper GnRH secretion[1]. In line with these observations, both male and female *Sema6a*[-/-] mice showed delayed puberty onset and sexual maturation defects (Fig. 4). The decrease in GnRH neuron innervation of the ME was not accompanied by a reduction in the number of GnRH neurons reaching the MPOA (Fig. 2). This excludes a decrease in neuronal number as the cause for the reduced density of GnRH-positive axons in the ME. Moreover, our data also argue against a direct effect of SEMA6A on GnRH neuron axons, as no expression of the SEMA6A receptors Plexin-A2 and Plexin-A4 was detected in MPOA-resident GnRH neurons (Supplementary Fig. 2).

The expression of *SEMA6A* in the external portion of the ME adjacent to fenestrated blood capillary loops invading the brain parenchyma, suggested a possible role in the NGE unit. This is a fundamental structure formed by a dynamic network of capillaries, β-tanycytes and neuroendocrine projections[5,39]. Interestingly, previous studies have described roles for semaphorins, such as *SEMA3A* and *SEMA7A*, in the control of GnRH neuron plasticity by modulating GnRH neurite extension[15] or tanycyte endfeet sprouting[17], respectively. However, evidence for direct control of ME vascular permeability by

semaphorins had not been reported. Here, by combining in vitro vascular permeability experiments and in vivo functional studies, we unveiled a thus far undiscovered role for *SEMA6A* in ME vascular barrier homeostasis. Specifically, we found that a secreted form of SEMA6A increases permeability of cultured EC monolayers (Fig. 6 and Supplementary Fig. 5), and that *Sema6a*[-/-] mice display a less permissive ME vascular barrier. This was associated with fewer fenestrated loop capillaries, but normal β-tanycyte morphology (Fig. 7 and Supplementary Fig. 6). These findings strongly support a direct role for SEMA6A in the regulation of ME vascular permeability.

The effect of SEMA6A on ECs is likely to be mediated by its canonical receptor Plexin-A2. Plexin-A2 is expressed by fenestrated capillaries of the ME in vivo (Fig. 7) and *PLXNA2* knockdown attenuated SEMA6A-induced vascular permeability in vitro (Fig. 6). These data show that SEMA6A can act via Plexin-A2 to modulate vascular permeability. Still, we cannot exclude that SEMA6A also interferes with Vascular endothelial growth factor receptor 2 (VEGFR-2) signaling, as previously reported in HUVECs[29,63], possibly through the binding with other SEMA6A interactors such as members of the Ena/VASP protein family[33,64]. VEGFR-2 is expressed by fenestrated ECs and has previously been shown to sustain ME vascular permeability through VEGF-A signaling[57].

Our findings provide support for a key role of SEMA6A in the regulation of NGE unit. The NGE unit regulates the outflux of neurohormones, including GnRH, to target tissues, as well as the influx of molecules from peripheral tissues. Therefore the ability of SEMA6A to regulate ME vascular barrier could modulate trafficking of molecules leading to changes in factor secretion or in uptake of circulating peptides[39,65]. In particular, decreased ME vascular permeability in the absence of SEMA6A (Fig. 7) may reduce the uptake of circulating factors that regulate GnRH neurite extension and/or GnRH secretion. For example, blood-borne Insulin-like growth factor 1 (IGF-1) and Fibroblast growth factor 21 (FGF-21) can induce the release of GnRH[66] and stimulate GnRH neurite outgrowth with subsequent peptide release[40], respectively (Fig. 9). Our observation that loss of SEMA6A reduces the diffusion of EB dye from the periphery through the ME barrier supports this model. Of note, mice lacking IGF-1 or FGF-21 receptors also exhibit DP phenotypes like *Sema6a*[-/-] mice[40,67].

The identification of a predicted pathogenic variant in *SEMA6A* in 6 individuals with a DP phenotype serves to translate our experimental observations to the clinic and provides a mechanistic link between SEMA6A deficiency and GnRH functionality in humans. DP is a highly heritable condition affecting up to 2% of the population[8,68]. Yet, despite recent progress, the genetic determinants of DP remain largely unknown[10]. The analysis of exome sequencing from 100 probands from a self-limited DP cohort led to the identification of this novel missense variant in *SEMA6A*, segregating with the DP trait in 4 individuals from the same family and two further probands from unrelated families. Interestingly, our patients do not exhibit other pituitary hormone deficiencies, strongly supporting a selective effect of SEMA6A on the regulation of the HPG axis. Our work provides strong support for a deleterious effect of this *SEMA6A*[I423T] variant on protein stability and localization. The DP patients in our cohort are heterozygous carriers of the *SEMA6A*[I423T] variant, whilst in the mouse model *Sema6a*[-/-] but not *Sema6a*[+/-] mice displayed DP. However, previous work on several related genes has demonstrated that even when patients carry heterozygous pathogenic mutants, only full knock-out, but not heterozygous, mice targeting the corresponding genes exhibit a pathological reproductive phenotype (e.g., *Sema3a*, *Sema7a*, *Plxna1*)[11,14,69]. The same phenomenon has been demonstrated for other developmental genes[70]. This would suggest that patient-derived mutations can exhibit a dominant effect on protein function, such as the altered stability and localization of the *SEMA6A*[I423T]

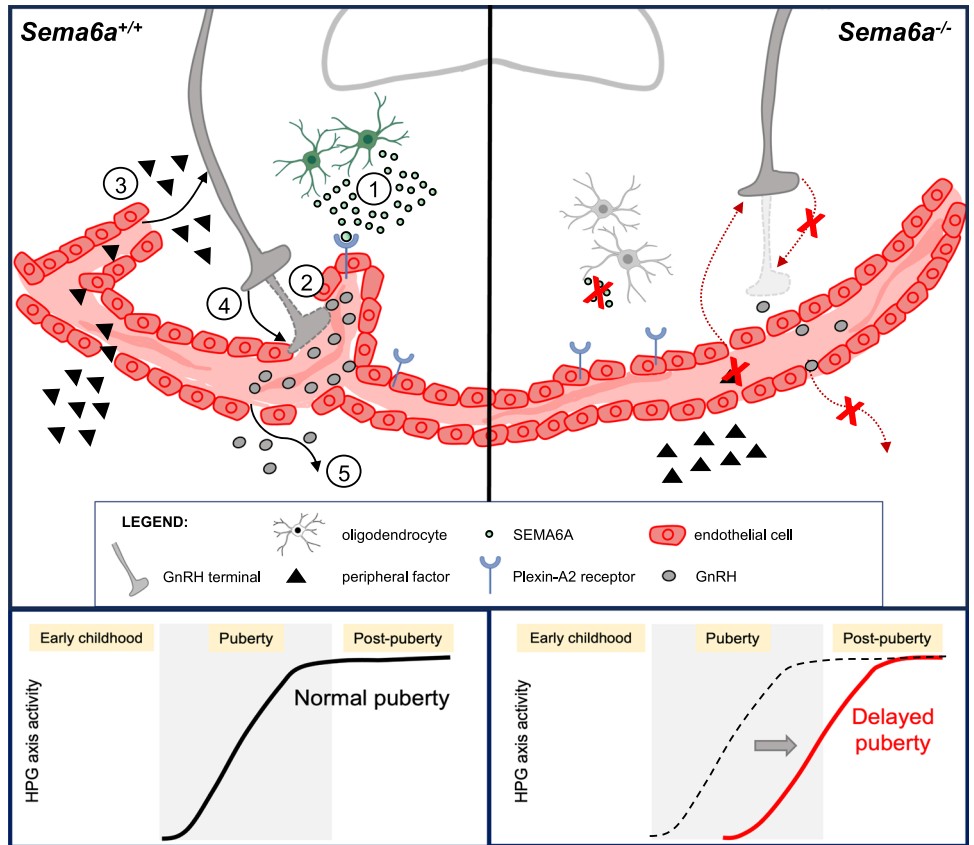

**Fig. 9 | SEMA6A modulates GnRH neuron homeostasis by tuning ME vascular permeability.** Schematic drawing representing the proposed cellular mechanisms through which SEMA6A is expressed by OLs to modulate GnRH homeostasis. In the ME of WT mice, SEMA6A (green dots) is produced by OLs (green cells) and distributed close to fenestrated ECs (red cells). In our model, SEMA6A acts on fenestrated ECs expressing Plexin-A2 (blue receptors) to increase the number of capillary loops in the brain parenchyma and maintain a permissive vascular barrier (1). Such permissive vascular barrier may ensure optimal GnRH peptide (gray dots) release in ME blood vessels either directly (2) or indirectly, via entry of molecules from the periphery (e.g., IGF-1, FGF-21; black triangles) in the ME (3) where they can act on GnRH neurons (in gray), by modulating the extension of axon terminals towards ME and ECs (4). This mechanism will ensure a normal release of GnRH peptide (5) (gray dots) and therefore a physiological puberty onset. Conversely, in *Sema6a*[-/-] mice, the lack of SEMA6A production by OLs (gray cells) in the basal part of the ME leads to fewer capillary loops and reduced vascular permeability. Consequently, direct GnRH release to the pituitary or entry of peripheral signals that normally sustain axon terminal elongation and GnRH secretion do not occur, resulting in a less innervated ME and ultimately in delayed puberty onset. Created with BioRender.com.

variant. In contrast, gene targeting strategies in mice do not usually affect the stability of the protein encoded by the *WT* allele in heterozygous animals.

In conclusion, our results uncover an additional role for the axon guidance cue SEMA6A in the control of puberty onset and sexual maturation and help to understand the pathogenic mechanisms underlying a novel *SEMA6A* variant in patients affected by DP. Specifically, these data unveil an additional mechanism through which OL-produced SEMA6A acts on Plexin-A2[+] ECs to regulate ME homeostasis and, as a consequence, GnRH neuron innervation. Our findings provide a framework for further understanding the genetic and molecular mechanisms that underlie puberty onset as well as for guiding diagnostic and therapeutic approaches in pubertal delay. Further, the outlined mechanism of disrupted ME vascular permeability could potentially have broader implications on various neuroendocrine functions beyond pubertal maturation.

## Methods
### Ethics
The human fetal tissue was provided by the Joint MRC/Wellcome Trust (grant #099175/Z/12/Z) Human Developmental Biology Resource (www.hdbr.org), with appropriate maternal written informed consent and approval from the Ethics Committee NRES Committee London – Fulham. HDBR is regulated by the UK Human Tissue Authority (HTA; www.hta.gov.uk) and operates in accordance with the relevant HTA Codes of Practice. Fetal tissues from both sources were obtained from terminations at 7 post conception weeks (Carnegie Stage 19). The human study protocol was approved by the Ethics Committee for Pediatrics, Adolescent Medicine and Psychiatry, Hospital District of Helsinki and Uusimaa (570/E7/2003). Ethical approval for human studies was granted by the UK London-Chelsea NRES committee (13/LO/0257). All participants provided written informed consent prior to study participation. The study was conducted in accordance with the guidelines of The Declaration of Helsinki. Participants did not receive compensation.

### Animals and tissue preparation
*HA-Sema6a*[fl/fl] mice, which harbor a 3xHA tag in exon 2 of *Sema6a* and lox sites surrounding exon 3, were a kind gift of Alex Kolodkin (Johns Hopkins University School of Medicine, Baltimore, USA). Germline deletion of *Sema6a* exon 3 was performed by crossing these mice with *EIIa-Cre* mice (JAX stock #003724)[71] to create *Sema6a*-null mice (here referred to as *Sema6a*[-/-]) that were maintained on a *C57Bl/6J* background (JAX strain, strain code 632, Charles River). Genotyping was performed by PCR using DNA from mouse tail or ear tissue as a template and primers flanking the HA tag sequence (fw '5-

CCTTGTCACACATGCAGTTG-3' and rev '5-TGAAGTAGAGAAGCA-3'), and by primers controlling for germline deletion of exon 3 (fw '5-GCTCTTTGGCAGGGGTGAAGTA-3' and rev '5-GTCCAAGTCCACGT-GACCC-3'). For all experiments, mice were housed socially with chow diet (CRM (E) SQC 811437, Special Diet Services) and water ad libitum, on a 12 h light/dark cycle, humidity of 45–65% and a temperature between 20 and 24 °C. The morning of vaginal plug observation was defined as embryonic day 0.5 (E0.5). The day of birth was defined as postnatal day 0 (P0). Timed pregnant females were sacrificed by cervical dislocation to collect embryos at E12.5, E14.5 and E18.5. Embryo heads were fixed for 3–6 h in 4% PFA at 4 °C. Adult mice were euthanized by overdosing Euthanimal (200 mg/mL, Alfasan) intra-peritoneal (i.p.), and transcardially perfused with ice-cold 4% PFA. Subsequently, brains were isolated and post-fixed in 4% PFA overnight at 4 °C. All samples for cryosectioning were cryoprotected overnight in 30% sucrose prior to OCT-embedding, whereas samples for paraffin embedding were dehydrated by immersion through crescent alcohol series and xylene. All in vivo experiments were conducted by certified personnel in compliance with the Animal Ethics Committee of Utrecht University (Dierexperimenten Ethische Commissie) (CCD licence: AVD115002016532 and AVD11500202114777) and the Animal Welfare Body of the University of Milan, and in agreement with Dutch laws (Wet op de Dierproeven, 1996; revised 2014), the Italian Minister of Health and European regulations (Guideline 86/609/EEC; Directive 2010/63/EU). Animal welfare was checked daily.

## Immunofluorescence labeling

20 μm cryostat sections of PFA-fixed embryos were incubated with serum-free protein block (DAKO) after permeabilization of sections with 0.1% TritonX-100. We used as primary antibodies mouse anti-nTUBB3 (1:500, clone Tuj1; Covance, cat. MMS-435P), rabbit anti-Peripherin (1:100; Millipore, cat. AB1530), rabbit anti-GnRH, (1:400; Immunostar, cat. 20075)[14], rabbit anti-BLBP (1:200; Millipore, cat. ABN14)[72], goat anti-SEMA6A (1:200; R&D Systems, cat. AF1615)[29], rabbit anti-GFAP (1:500; Dako, cat. Z0334)[73], rabbit anti-Synaptophysin (1:200; Synaptic Systems, cat. 101002), mouse anti-Vimentin (1:200; Santa Cruz Biotechnology, cat. sc-373717)[74], rabbit anti-Oligo2 (1:300; Millipore, cat. AB9610)[75], rabbit anti-Gst P1 (1:200; MBL, cat. 312)[76], rat anti-PDGFR-α (1:100, clone APA5; Invitrogen, cat. 14-1401-82)[77], goat anti-Plexin-A2 (R&D Systems, cat. AF5486)[78], rabbit anti-Plexin-A4 (Novus Biologicals, cat. NBP1-85128)[79], mouse anti-Plexin-A2 (1:100, clone A-2; Santa Cruz Biotechnology, cat. sc-393939), rabbit anti-PECAM1 (1:200; Abcam, cat. ab28364)[80] and rat anti-PECAM1 (1:200; BioLegend, cat. 102502). To visualize GnRH neurons in the MPOA when another antibody raised in rabbit was used, we employed goat anti-Plexin-D1 antibody (1:200; R&D Systems, AF4160), as Plexin-D1 can be used as marker for MPOA-resident GnRH neurons[81]. For blood capillary loop analysis, 16 μm cryostat sections of PFA-fixed brains at ME level were first boiled in an antigen retrieval solution (0.1 M citric acid, 1 M sodium citrate) for 20 min and then incubated in a blocking solution (PBS 0.1% TritonX-100, 0.05% Tween-20, 2.5% donkey serum, 5% BSA) and primary rat anti-PLVAP antibody (1:100, clone Meca32; BD Biosciences, cat. 550563)[58]. Secondary antibodies used were Alexa Fluor 488-, Cy3- or Alexa Fluor 647-conjugated donkey anti-goat, anti-mouse, anti-rabbit and anti-rat Fab fragments (1:200; Jackson Immunoresearch). In some experiments, blood vessels were labeled using biotinylated IB4 (1:100, Vector laboratories) followed by Cy3-conjugated streptavidin. Nuclei were counterstained with DAPI (1:10000; Sigma Aldrich).

## Immunohistochemistry

20 μm cryostat sections of PFA-fixed samples were incubated with hydrogen peroxide to quench endogenous peroxidase activity, and sequentially incubated with 10% normal goat serum in PBS or serum-free blocking solution (Agilent, cat. X0909) and then immunostained with rabbit anti-GnRH (1:1000; Immunostar), previously validated to recognize both the pre-hormone and the processed hormone[82]. 8 μm microtome sections of PFA-fixed paraffin-embedded testes were similarly processed and incubated with rabbit anti-CYP17A1 antibody (1:200; Proteintech, cat. 14447-1-AP)[18]. Sections were then incubated with anti-rabbit biotinylated antibody (1:400; Vector Laboratories) and developed with the ABC kit (Vector Laboratories) and 3,3-diaminobenzidine (DAB; Sigma). To determine the total number of GnRH neurons at E14.5, E18.5 and postnatal stages, 20 μm coronal sections through each entire head (E14.5), FB (E18.5) or MPOA (postnatal) were immunolabelled for GnRH and all GnRH+ cells were manually quantified upon image acquisition[14]. To compare the abundance of GnRH-positive neurites in the ME, we measured the pixel intensity of GnRH staining using ImageJ software (v1.52a, NIH)[14]. To compare the abundance of Leydig cells in testes, we quantified for each section the percentage of CYP17A1+ area using ImageJ software (v1.52a, NIH). 10 μm paraffin-embedded human embryo sections were deparaffinized in xylene and rehydrated in decrescent ethanol series before antigen retrieval in 10 mM sodium citrate pH 6.0; after blocking, sections were incubated with goat anti-mouse SEMA6A (1:200; R&D Systems, cat. AF1615) and anti-goat biotinylated antibody (1:400; Vector Laboratories) and developed as described above.

## Hematoxylin & eosin staining and follicle quantification

8 μm microtome sections of PFA-fixed paraffin-embedded ovaries were rehydrated with xylene/decrescent alcohol series, stained with Mayer's hematoxylin (Bio-Optica) for 2 min and Eosin Y alcoholic solution (Bio-Optica) for 3 min and finally dehydrated with crescent alcohol series/xylene. Two sections for each female mouse were considered for the count of ovarian follicles. Follicles were distinguished into primary (single layer surrounding the oocyte), secondary (two layers surrounding the oocyte), pre-antral (three or more layers surrounding the oocyte) and early antral (three or more layers surrounding the oocyte with the presence of a cavity) and quantified[83].

## Determination of male balanopreputial separation, female vaginal opening and first estrous

*Sema6a*[-/-], *Sema6a*[+/-], and *Sema6a*[+/+] male and female mice were weighed and checked daily for BPS and VO[37,38,84], respectively, after weaning from P22 onwards. When female mice showed VO, vaginal smears were obtained every 24 h to assess the start of the estrous cycle by gently pipetting and resuspending 30 μL saline solution (0.9% NaCl) in the VO using a P200 pipette. The collected cells were transferred to a glass microscope slide and air-dried. Slides were then treated with 70% EtOH for 3 min, followed by 0.1% Cresyl Violet stain in 50% EtOH for 1 min, and briefly washed with 100% EtOH. Subsequently, smears were evaluated using light microscopy and the relative amount of nucleated, anucleated epithelial cells and leukocytes present was used to determine the phase of the estrous cycle (Supplementary Fig. 3c)[37]. Gross absence of leukocytes and presence of multitudes of anucleated epithelial cells was considered as estrous. Tissues for analysis were collected from male and female mice at the day of BPS and FE, respectively. Mice were euthanized by cervical dislocation, brains and gonads were isolated and fixed overnight in 4% PFA at 4 °C, and pituitaries were snap-frozen and stored at −80 °C. Males were collected from 13 litters, females were collected from 14 litters.

## Evans Blue (EB) in vivo permeability assay

At P30, *Sema6a*[-/-], *Sema6a*[+/-], and *Sema6a*[+/+] female mice were euthanized by overdose of Euthanimal i.p. (200 mg/mL, Alfasan). Subsequently, animals were transcardially perfused with 20 mL sterile-filtered 68 mg/mL EB dye (Sigma-Aldrich) dissolved in 4% BSA in PBS,

followed by perfusion with 25 mL PBS. Perfusion speed was kept at 3 mL/min. Following perfusion, brains were isolated and cryoprotected by incubation in ascending concentrations of sucrose (10%–20%–30%) in PBS for 30 min each at 4 °C. Brains were then embedded in OCT and snap-frozen in dry-ice. *Sema6a*⁺/⁻ and *Sema6a*⁺/⁺ females who did not yet show VO at P30 were excluded from analysis. *WT* and *Sema6a* mutant animals were collected from 4 litters. For analysis, 30 μm cryosections were directly imaged under laser scanning confocal microscope to visualize EB staining excited at 640 nm. EB diffusion in the ME parenchyma was assessed by quantifying the background-normalized mean gray values of three equally sized fields per section, randomly placed in the ME region, using ImageJ software (v1.52a, NIH).

### RNA extraction and RT-qPCR
Total RNA was obtained from HUVEC cell pellets and mouse pituitaries using Trizol/NucleoZOL (Macherey-Nagel). 500 ng of RNA were then retrotranscribed using All-in-one 5x RT MasterMix kit (Abm, cat. G592) and random primers. RT-qPCR was performed on CFXOplus96 thermocycler (Bio-Rad) using Master Mix with SYBR Green (Takara Bio) and specific primers (mouse *Lhb*: fw 5′-CAAGAATGGAGAGGCTCCAG-3′, rev 5′-ACTGGGCAGAACTCATTCTCTG-3′; mouse *Gapdh*: fw 5′-CATCCCA-GAGCTGAACG-3′, rev 5′-CTGGTCCTCAGTGTAGCC-3′; human *PLXNA1*: fw 5′-GTGGAGAGGTACTATGCAGAC-3′, rev 5′-CATGCTGTTGAA CTGGC TC-3′; human *PLXNA2*: fw 5′-AAGTATAGTGAGGAGCTCATCG-3′, rev 5′-CTGCTCCACCTTATAAGCCA-3′; human *PLXNA3*: fw 5′-GTGCCAG TTTGTAAGGAGAG-3′, rev 5′-CAATGGCAAGTGTGATGGG-3′; human *PLXNA4*: fw 5′-AATGACCGCATTAAGGAGC-3′, rev 5′-AATGGTTAA-GAGCGCACTG-3′; human *RPLP0*: fw 5′-CAGATTGGCTACCCAACTGTT-3′, rev 5′-GGGAAGGTGTAATCCGTCTCC-3′). The quantification cycle (ΔCq) value and the ΔΔCq were calculated relative to control samples using Cq threshold values normalized to the housekeeping gene *RPLP0*/*Gapdh*.

### scRNA-seq analysis
Raw data for adult brain scRNA-seq datasets from Tabula Muris and EC atlas were obtained from NCBI gene Expression Omnibus (GSE109774) and EBI ArrayExpress (E-MTAB-8077), respectively, and analyzed with RStudio v.1.3.1056. The raw gene expression matrices (UMI counts per gene per cell) were filtered, normalized and clustered using the R package Seurat v.3.2.0. Cells containing less than 200 feature counts and genes detected in less than 3 cells were removed. Downstream analysis included data normalization ("LogNormalize" method and scale factor of 10,000) and variable gene detection ("vst" selection method, returning 2000 features per dataset). The principal components analysis (PCA) was performed on variable genes, and the optimal number of principal components, PCs, for each sample was chosen using the elbow plot. The selected PCs were used for Louvain graph-based clustering at a resolution of 0.3 followed by manual curation based on marker genes to obtain the final seven and four clusters from Tabula Muris and EC atlas brain datasets, respectively. Uniform manifold approximation and projection (UMAP) was chosen as a non-linear dimensionality reduction method, and each relevant gene was then examined using the *FeaturePlot* and *VlnPlot* functions. Cluster cell identity was assigned by manual annotation based on known marker genes. Clustering was then manually curated based on marker genes to obtain the final seven and four clusters from Tabula Muris and EC atlas brain datasets, respectively, followed by subset selection in the Tabula Muris dataset of the cluster containing *Emcn*⁺ ECs.

### Cell culture, transfection and lentivirus infection
COS-7 cells (American Type Culture Collection, cat. CRL-1651) were grown as a monolayer at 37 °C in a humidified CO₂ incubator in complete DMEM (Euroclone) and supplemented with 10% fetal bovine serum (FBS; Gibco). Subconfluent cells were harvested by trypsinization and cultured in 57 cm² dishes. For transfection, COS-7 cells (at 80% confluence) were grown in culture plates in complete culture medium for 24 h and incubated for 48 h with the selected expression vector (1 μg/mL) in the presence of Lipofectamine 3000 (Invitrogen) according to the manufacturer's instructions. To visualize Endoplasmic Reticulum (ER), the mEmerald-ER-3 vector (gift from Michael Davidson; Addgene plasmid #54082) was co-transfected with the *SEMA6A*ᴵ⁴²³ᵀ vector (ratio 1:4). The conditioned media of COS-7 cells either mock transfected or expressing SEMA6Aᴱᶜᵀᴼ were collected after 48 h and stored at −80 °C. HUVECs from pooled donors (Promocell, cat. FB60C12203) were cultured on fibronectin (Promocell) or rat collagen I coated vessels (Corning, cat. DLW354236) in EGM-2 complete medium (Promocell). To generate stable *PLXNA2* knockdown cells, HUVECs were separately infected with a set of lentiviral particles expressing four different *sh*RNAs (A-D) against human *PLXNA2* (Origene, cat. TL302405V) with a calculated MOI of 10; cells infected with lentiviral particles expressing scramble *sh*RNA were used as controls. At 48 h post infection, cells were selected with 1 μg/mL puromycin. Knockdown efficiency was evaluated by RT-qPCR.

### Plasmids
The *pAPtag-5* vector (6.6 kb) (GenHunter, cat. Q202) was chosen for the cloning of the shorter isoform of human *SEMA6A* gene (NM_020796.5), encoding for a protein of 1030 amino acids (expected molecular weight 120 kDa). Two different vectors have been prepared with human *SEMA6A*ᵂᵀ and *SEMA6A*ᴵ⁴²³ᵀ genes inserted between the NheI and the XhoI restriction enzyme sites, in frame with c-myc tag, cutting off the AP protein. The empty vector has been used as a control vector. The *pCAGG-AP-SEMA6A-ecto-Fc-His* vector was employed to produce mouse SEMA6A soluble extracellular domain ranging from amino acid 18 to 688 (SEMA6Aᴱᶜᵀᴼ)[31].

### Isolation and culture of ECs from adult mouse brain
mBECs were isolated from 3 to 5 brains of adult WT *C57Bl/6* mice by slightly modifying an existing protocol[52]. Briefly, brains were cut along the longitudinal fissure to separate the two hemispheres and the cerebellum was removed from the rest of the brain. The meninges were removed by rolling the individual parts on 3 M paper. Brain hemispheres were cut and minced using a scalpel, resuspended in dissection buffer (Optimem, 2% FCS, 2 ng/mL heparin and pen/strep), and passed through a 10 mL serological pipet until no evident clumps were present. The homogenate was centrifuged (300 g, 5 min) and the pellet was digested in dissection buffer containing 1 mg/mL of collagenase/dispase (Sigma Aldrich) and 50 μg/mL of DNase (Sigma Aldrich) on a benchtop shaker (900 RPM) for 45 min at 37 °C. To remove the myelin, the pellet was resuspended in 22% (w/v) of bovine serum albumin (BSA) in PBS and centrifuged (1620 g for 15 min at 4 °C) for 3 times. The combined pellets were then resuspended in EGM complete medium (Promocell) supplemented with 5 μg/mL of puromycin to kill non-ECs and seeded on collagen coated (rat tail collagen I 50 μg/mL in PBS; Corning, cat DLW354236) 0.4 μm pore-size Transwell Permeable Supports (Merck, cat. MHCT24H48). Culture medium was replaced after 48 h with complete medium and changed every other day until confluence. mBECs purity was evaluated by morphological inspection and immunofluorescence staining for the EC markers PECAM1 and Cadherin-5 as described below.

### Immunofluorescence
Cells were fixed with 4% paraformaldehyde (PFA) fixative solution for 15 min at room temperature and washed with PBS. COS-7 cells were blocked and permeabilized with 10% goat serum 0.1% TritonX-100 (all from Sigma Aldrich) solution and incubated with mouse anti c-myc

(1:500, clone 9E10; ThermoFisher, cat. 13–2500)[18] antibody overnight at 4 °C. mBECs were blocked and permeabilized with 5% donkey serum, 5% BSA, 0.1% Triton X-100 (all from Sigma Aldrich) solution for 30 min at room temperature. After blocking, mBECs were incubated with goat anti-Cadherin-5 (1:200; Santa Cruz Biotechnology, cat. sc-6458), rat anti-PECAM1 (1:200; BioLegend, cat. 102502)[85], rat anti-ICAM-2 (1:200; clone 3C4; BD Biosciences, cat. 553326)[55] and goat anti-Plexin-A2 (1:200; R&D Systems, cat. AF5486) antibodies overnight at 4 °C. Secondary antibodies used were Alexa Fluor 488-, Cy3 or Alexa Fluor 647-conjugated donkey anti-goat, anti-mouse and anti-rat Fab fragments (1:200; Jackson Immunoresearch). F-actin staining was performed incubating cells with phalloidin-TRITC (1:400; Sigma Aldrich, cat. P1951) for 30 min at 37 °C[86]. Nuclei were counterstained with DAPI (1:10000; Sigma Aldrich).

### ECs permeability assay
HUVECs or mBECs were seeded on fibronectin or collagen I coated 0.4 µm pore-size Transwell Permeable Supports (Merck, cat. MHCT24H48), cultured in complete culture EGM-2 or EGM medium (Promocell), respectively, and let to form a monolayer. After the establishment of a stable monolayer, cells were starved overnight with DMEM without phenol red supplemented with 2% FBS and antibiotics. The next day, HUVECs or mBECs were stimulated with conditioned medium containing murine SEMA6A$^{ECTO}$ or a mock control added in the apical or basal side of the transwell. DMEM without phenol red supplemented with 2% FBS with and without VEGF-A (50 ng/mL) was used as negative and positive control, respectively. TEER was assessed using a Millicall-ERS Voltohmeter (World Precision Instruments)[87,88]. TEER measurements were performed using the same conditions to allow comparison of different groups and calculated as area-integrated resistance ($\Omega cm^2$). The percentage of TEER change was calculated with respect to time 0 for each indicated time point.

### Human genetic analysis
Patients from a large Finnish cohort with self-limited DP were analyzed for this study. Patients referred with self-limited DP to specialist pediatric care in central and southern Finland between 1982 and 2004 were identified. All participants ($n = 492$) met the diagnostic criteria for self-limited DP[89]. Inclusion criteria were the onset of Tanner genital stage II (testicular volume > 3 mL) > 13.5 years in boys or Tanner breast stage II > 13.0 years in girls (i.e., two SD later than average pubertal development) and their unaffected relatives. Normal values of thyroid function tests (TFTs) and IGF-1 for bone age were also used as inclusion criteria. Chronic illness as a cause for functional HH was excluded by medical history, clinical examination, and biochemical investigations. Congenital or acquired HH, if suspected, was excluded by spontaneous pubertal development by 18 years of age at follow-up. Sex was considered in the study design, and sex of participants was determined based on self-report.

### Whole exome sequencing analysis
Whole exome sequencing was performed on DNA extracted from peripheral blood leukocytes of 100 probands, using an Agilent V5 platform and Illumina HiSeq 2000 sequencing. The exome sequences were aligned to the UCSC hg19 reference genome using the Burrows-Wheeler Aligner software (BWA-MEM [bwa-0.7.12]). Picard tools (picard-tools-1.119) was used to sort alignments and mark PCR duplicates. The genome analysis toolkit (GATK-3.4–46) was used to realign around indels and recalibrate quality scores using dbSNP, Mills and 1000 genomes as reference resources. Variant calling and joint genotyping using pedigree information was performed using Haplotype-Caller in GVCF mode from the genome analysis toolkit. The resulting variants were filtered using the variant quality score recalibration (VQSR) function from GATK. An analysis of the called variants was performed using Ingenuity Variant Analysis (QIAGEN Redwood City, www.qiagen.com/ingenuity). Filtering for potential causal variants was carried out using filters for quality control (read depth and Phred strand bias), minor allele frequency (MAF < 0.5% in the ExAC and Genome Aggregation Database (gnomAD) databases v2.0.2), predicted functional annotation (Poly-Phen, SIFT, REVEL, CADD score) and conservation score (GERP). Potential causal variants were confirmed by Sanger sequencing.

### In silico mutagenesis
The CryoEM structure of the SEMA6A homodimer in complex with *C. sordellii* lethal toxin TcsL (PDB ID: 6WTS; https://doi.org/10.2210/pdb6WTS/pdb)[59] was retrieved from the RCSB Protein Data Bank. After a structure preparation procedure via Schrödinger BioLuminate Protein Preparation Tool, including an energy minimization using the OPLS3e force field[90], the TcsL toxin was removed. The evaluation of the impact of I423T mutation on the protein stability has been carried out with the Schrödinger BioLuminate Residue Scanning Tool. Stability is computed from a thermodynamic cycle which considers the relative stability of each entity: unfolded SEMA6A, folded SEMA6A, unfolded mutated SEMA6A and folded mutated SEMA6A[91].

### Immunoblotting
Transfected COS-7 cells were lysed in 150 mM NaCl, 50 mM Tris-HCl (pH 7.4), and 1% TritonX-100, supplemented with protease and phosphatase inhibitors (Roche). Lysates were centrifuged at 10,000 g for 10 min at 4 °C and protein concentration determined with the Bradford assay (Bio-Rad). Protein lysates (20 µg) were used for SDS-PAGE. Proteins were transferred to nitrocellulose membranes (Bio-Rad) and immunoblotted with mouse anti c-myc (1:500, clone 9E10; ThermoFisher, cat. 13–2500) or rabbit anti-GAPDH (1:1000; Cell Signaling, cat. 5174S) followed by goat anti-mouse (1:4000; Agilent, cat. P044701-2) or anti-rabbit (1:2000; Sigma-Aldrich, cat. A4914) HRP-conjugated antibodies. Images were acquired by using Image Lab software (v5.2.1, Biorad). For quantification, 3 independent experiments were performed for each condition, the OD of the signal measured with ImageJ software (v1.52a, NIH) and the mean pixel intensity calculated.

### Image acquisition
Brightfield images were acquired with IS Capture software (v3.6.7, TiEsseLab). Cells and mouse tissues were examined with a Nikon A1R/AX laser scanning confocal microscope equipped with a Nikon A1/AX plus camera and the following objectives (Nikon): Plan Fluor 10X DIC L N1 (NA 0.3), Plan Fluor 20X DIC N2 (NA 0.5), 20X Plan Apo λD OFN25 DIC N2 (NA 0.80), 40X Apo LWD WI λS DIC N2 (NA 1.15), 60X Plan Apo λD OFN25 DIC N2 (NA 1.42). DAPI, Alexa 488, Cy3 and Alexa 647 were excited at 405, 488, 561 nm and 640 nm and observed at 425–475, 500–550, 570–620 and 663–738 nm, respectively. Plexin-A2 polarization and EB in vivo permeability assays were examined with a Zeiss LSM900 Airyscan laser scanning confocal microscope equipped with a Zeiss Axiocam 305 camera and the following objectives (Zeiss): EC Plan-Neofluar 20X M-27 (NA 0.5), 63X Plan-Apochromat Oil DIC M27 (NA 1.40). DAPI, Alexa 488, Cy3 and Alexa 647 were excited at 405, 488, 561 nm and 640 nm and observed at 400–605, 410–545, 535–617 nm and 645–700 nm. 1024 × 1024 pixels images were captured in a stepwise fashion over a defined z-focus range corresponding to all visible fluorescence within the sample. Maximum projections of the z-stack were performed post-acquisition by using NIS Elements AR (v5.21.03, Nikon) or ZEN 3.0 Suite (Zeiss) software. Adobe Photoshop CS6 (Adobe) and Fiji (v2.3.0/1.53 f, NIH) software were used to obtain the presented images.

## Statistics

Statistical tests employed and number of samples analyzed are outlined in figure legends and were conducted when the experiment had been performed a minimum of 3 times on a minimum of 3 distinct samples, using Prism software (v8.2.1, GraphPad). Expression studies has been performed a minimum of 2 independent times. Data are presented as mean ± SD; the differences between means are reported as mean difference ± SEM. Results are considered significant with a $p$ value less than 0.05.

## Reporting summary

Further information on research design is available in the Nature Portfolio Reporting Summary linked to this article.

## Data availability

All data supporting the findings described in this manuscript are available in the article and its Supplementary Information files. Source data are provided with this paper. All unique materials are readily available from the authors upon request. The exome sequencing data are available under restricted access due to ethics restrictions on storing and sharing our pediatric patient exome data. These data will have controlled access and will be limited to individuals who enter a research agreement; use of these genomic data will be restricted to those named on the agreement, and exome data will be patient de-identified. Data will be shared after contacting the corresponding author (A.C.: anna.cariboni@unimi.it), who ensures a response and the access to the data within one month. The SEMA6A variant has been deposited in the ClinVar database under accession code VCV002571587.2. The single-cell transcriptomic data Tabula Muris (https://tabula-muris.ds.czbiohub.org/)[46] and EC atlas (https://endotheliomics.shinyapps.io/ec_atlas/)[53] used in this study are available in the GEO and EBI ArrayExpress database under accession code GSE109774 E-MTAB-8077. The bulk RNA-seq expression data used in this study were analyzed through BulkECexplorer (https://ruhrberglab.shinyapps.io/BulkECexplorer/)[54], an online resource that surveys 240 publicly available bulk RNA-seq datasets from five human and mouse EC subtypes, including HUVECs and mBECs. In addition, the following databases were used: gnomAD (https://gnomad.broadinstitute.org/), VarSome (https://varsome.com) and PDB (https://www.rcsb.org). Source data are provided with this paper.

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

## Acknowledgements

This work received support from: the Italian Ministry of Health (GR-2016-02362389) and the Italian Fondazione Telethon (GMR22T1054) to A.C.; the Netherlands Organization for Scientific Research (ENW-VICI) to R.J.P.; the National Institute for Health Research (CL-2017-19-002), Wellcome Trust (222049/Z/20/Z), Barts Charity [MGU0552] and the Rosetrees Trust [M222-F1] to S.R.H.; Fondazione Cariplo (2018-0298) and the Fondazione Italiana per la Ricerca sul Cancro (AIRC, 22905) to A.F. A.L. was partially supported by Fondazione Veronesi and R.O by Fondazione Collegio Ghislieri and Fondazione Telethon. The authors acknowledge support from the University of Milan, MIUR Progetto di Eccellenza and Piano Sostegno alla Ricerca UNIMI (linea 2 azione C) through the APC initiative. This research was funded in part by the Wellcome Trust [Grant number 222049/Z/20/Z]. For the purpose of open access, the authors have applied a CC BY public copyright licence to any Author Accepted Manuscript version arising from this submission. We would like to thank Chiara Parravicini, Silvia Manfro and Marta Frisa for technical help, Diego De Stefani for sharing the ER-vector, Filippo Casoni, Giulia Magni and Davide Marangon for sharing antibodies, Alex Kolodkin for sharing *HA-Sema6a*^fl/fl^ mice, Chiara Colletto for assistance with scRNA-seq analysis, Nicky van Kronenburg for help with mouse breeding and Christiana Ruhrberg and Sara Campinoti for help with human tissues. Part of this work was carried out at NOLIMITS, an advanced imaging facility established by the Università degli Studi di Milano. The human embryonic and fetal material was provided by the Joint MRC/Wellcome Trust (grant #099175/Z/12/Z) Human Developmental Biology Resource (www.hdbr.org). Some of the figures were prepared using Biorender.com (agreement #JF25TPXRCF).

## Author contributions

A.L. R.O., M.H.v.d.M., E.Y.v.B., M.G.V., S.R.H, A.F., R.J.P. and A.C. conceived and designed experiments, collected and analyzed data, and wrote the manuscript; A.L., R.O. and A.C. performed immunohistochemical experiments; M.H.v.d.M., E.Y.v.B., M.G.V. performed in vivo experiments; A.L., A.J.J.P., R.A., V.A. and F.A. performed in vitro experiments; S.R.H and L.D. obtained and analyzed patient data; I.E and L.P. performed in silico modeling; C.T and M.S. performed in vitro EC studies; R.J.P. and A.C. performed supervision and provided final approval of the manuscript.

## Competing interests

The authors declare no competing interests.
