## [Peer Review File · Nature Communications]

REVIEWER COMMENTS

Reviewer #1 (Remarks to the Author):

The manuscript 'SEMA6A drives GnRH neuron-dependent puberty onset by tuning median eminence vascular permeability' by Lettieri et al. describes the regulation of the onset of puberty by the axon guidance molecule Semaphorin6A (SEMA6A). The authors found that SEMA6AKO female mice expose a delayed puberty which was correlated with an effect of SEMA6A in median eminence (ME) vascular permeability. Moreover, they were able to identify a SEMA6A genetic variant in patients affected from delayed puberty. The paper is well written and nicely illustrated and the results are very interesting and evocative, however there are some aspects of the suggested mechanism that require further characterization prior acceptance for publication.

Please see below the specific points:

1. The authors assess by immunostaining the expression of SEMA6A along the development and migration of GnRH neurons. SEMA6A expression does not overlap with GnRH neuronal terminals neither with fenestrated vessels in the ME. It is important to determine which cells are expressing SEMA6A to better analyse the observed phenotype. It could be that the mechanism is not directly mediated by GnRH neurons.
2. In line with the previous point, the authors did not check the involvement of b2-tanycytes which are located in the basal part of the third ventricle and are in contact with blood vessels. Tanycytes have been involved in the ME permeability (see i.e. Langlet et al, Cell Metab, 2013), they express other semaphorins and mediate the interaction between GnRH axons and the vasculature (Parkash et al, Nat comm, 2015). Is SEMA6A expressed by the b2-tanycytes? Are the b2-tanycytes altered in the SEMA6A KO mice? Are there differences in the 3rd ventricle formation? (one could think so from the figure Fig. 2 M, N).
3. The TEER experiment in Fig 4F would suggest that the expression of SEMA6A receptors is polarized in endothelial cells, is it the case? Which Plexin receptors are participating in the permeability response in the endothelial cells? Can the silencing of the receptors rescue the increase of permeability? Is there expression of Plx2A and Plx4A in the ME? Also, Fig Suppl 2C and G need a positive control to be sure that the antibody used in the staining worked but the protein is absent in the region.
4. In relation of the point above, it would be important to show raw TEER values instead of relative TEER values versus T0 to better evaluate the tightness of the barrier. I would also suggest to show also a longer time period of TEER measurement to observe the maintenance of the effect overtime.
5. The authors use the expression of PLVAP as a proxy for vascular permeability (Fig 4), however this result needs a more convincing method to evaluate the leakage in the fenestrated vessels of the ME. The authors should use tracers of different size (such as cadaverine, dextran 10kDa or fluorescent albumin) to assess the changes in vascular permeability.

6. It would be also important to better characterize the vascularization defects in the ME in the SEMA6AKO at embryonic and adult time points in correlation with other defects observed in the mutant mice. For instance, are there only less loops or also less vessels? I would suggest to use other vascular markers, such as Glut1 or CD31, since they are more specific than IB4.

7. The mutation identified in human samples was found mostly in males. Do SEMA6A KO mouse males expose any sexual maturation defect?

Reviewer #2 (Remarks to the Author):

In this study, Lettieri et al. explore the roles of the member of the semaphorin family, SEMA6A, in the control of ME vascular permeability and thereby GnRH function, as putative component for pubertal control. The study is nicely presented and contains a substantial amount of experimental data, including preclinical and clinical findings, which collectively point to a previously undocumented role of SEMA6A in the control of GnRH system, by modulating GnRH innervation of the ME.

While the experimental data are convincing and the results are coherently discussed, there are several points that require further attention by the authors.

1. Admittedly, the role of SEMA6A in the control of GnRH neuronal system has not been previously reported. Yet, other members of the SEMA family, such as SEMA3A and SEMA7A, had been implicated previously in the control of different aspects of GnRH neurons. While the original papers referring to this regulatory role are quoted in the paper, discussion of similarities and differences between SEMA6A and these other members is relatively superficial and deserves more attention. Overall, it is crucial that the authors make a stronger case for the novelty and specificity of the mechanism proposed, as the reader might be left with the impression that SEMA6A is just another element of the complex regulatory pathway of SEMA factors involved in GnRH control, as previously documented.

2. The mechanism proposed is related with changes in vascular ME permeability, which may impair the inflow and outflow of multiple factors across the blood brain barrier at the level of the ME. If this is the case, are the authors sure the reported effects are specific of GnRH neurons, or might this be just one of the manifestations of a wider perturbation, as one might expect from such an alteration in the vascular permeability at the ME. The narrow focus on puberty and GnRH neurons might obscure identification of additional targets and mechanisms for the observed phenotype, or might prevent identification of other phenotypes related with impairment of SEMA6A. This needs to be discussed in deeper detail and specificity of the findings, ideally, further supported with additional data.

3. While considerable focus is placed on the role of SEMA6A on pubertal maturation, the indices used for monitoring puberty in the mouse model are rather limited, even if findings are very convincing. For instance, there is a clear delay in vaginal opening, but this phenomenon is not confronted with analyses of other, more direct parameters of pubertal activation, such as ovarian maturation, first estrus and/or changes in LH and/or estradiol levels. In the absence of such additional parameters, the argumentation of pubertal changes is weakened and the mechanisms underlying the observed pubertal delay remain superficially characterized. This by no means implies the findings are not convincing, but may fall short in proving the major conclusion, as vaginal opening might not sufficiently capture the complexity of pubertal maturation.

Reviewer #3 (Remarks to the Author):

In this study, Lettieri A et al. report a novel loss-of-function mutation in SEMA6A. SEMA6A can be a potential candidate gene for delayed puberty. This finding was confirmed in a validation cohort and supported by functional in vitro and in vivo studies. The study addresses an interesting topic on the pathogenesis of hypogonadism or delayed puberty.

Despite these strengths, there are some issues that need careful consideration and eventual revision, as they may weaken the conclusions and general impact of the study.

1. The most significant concern of these findings is that the missense variant of the SEMA6A gene is not constrained. The heterozygous variant in this study is unlikely to have a clinical impact.
2. In the animal functional study, heterozygous mutant mice showed no difference from wild-type mice in the vaginal opening. It would be good to add more explanation about this in the discussion section.
3. Did the authors confirm this novel variant by sanger sequencing?
4. Clinical information on patients is lacking. Are all patients "delayed puberty"? Clinical information such as the age of onset of puberty and hormone levels for the patient should be added to the table.
5. The use of COS-7 cells addresses this, but this cell line might not reflect well the GnRH neuronal physiology. I would suggest to confirm using GT1-7 cells.

Point-to-point rebuttal - SEMA6A drives GnRH neuron-dependent puberty onset by tuning median eminence vascular permeability

We would like to thank the reviewers for their careful and positive assessment of our work and for giving us the opportunity to further improve our study. We have revised the manuscript according to the reviewers' suggestions and have performed new mouse crosses, experiments, and analyses. As a result of additional experiments, several main and Supplementary Figures and Tables have been modified, and new data have been added to several main and Supplementary Figures. Finally, we have made additional textual changes to the manuscript reflecting points raised by the reviewers (in blue text in the main manuscript). Below we include a detailed response to the reviewers' comments (also in blue text):

Reviewer #1:

The manuscript 'SEMA6A drives GnRH neuron-dependent puberty onset by tuning median eminence vascular permeability' by Lettieri et al. describes the regulation of the onset of puberty by the axon guidance molecule Semaphorin6A (SEMA6A). The authors found that SEMA6AKO female mice expose a delayed puberty which was correlated with an effect of SEMA6A in median eminence (ME) vascular permeability. Moreover, they were able to identify a SEMA6A genetic variant in patients affected from delayed puberty. The paper is well written and nicely illustrated and the results are very interesting and evocative, however there are some aspects of the suggested mechanism that require further characterization prior acceptance for publication.

We thank the Reviewer for these positive comments.

Please see below the specific points:

1. The authors assess by immunostaining the expression of SEMA6A along the development and migration of GnRH neurons. SEMA6A expression does not overlap with GnRH neuronal terminals neither with fenestrated vessels in the ME. It is important to determine which cells are expressing SEMA6A to better analyse the observed phenotype. It could be that the mechanism is not directly mediated by GnRH neurons.

Reply: To address this point, we have extended the expression analysis to determine which cells are expressing SEMA6A in the median eminence, by performing double immunofluorescence experiments with specific neuronal and glial markers, including β -tancytes. Our results, now included in the revised manuscript at pages 10-11, show that SEMA6A is expressed by oligodendrocytes at specific stages of maturation. Specifically, we found that SEMA6A is expressed by pre-myelinating oligodendrocytes expressing Oligo2 and Gst P1, while SEMA6A is not expressed by oligodendrocyte progenitor cells expressing PDGFR- α (Figure 5e and Suppl. Figure 4a,b). Additional double immunofluorescence analysis with markers for neuroendocrine cells (synaptophysin), β -tancytes (vimentin) and astrocytes (GFAP), which represent other essential cell types known to modulate median eminence plasticity and GnRH secretion, did not show colocalization with SEMA6A (Figure 5a-d). Further, we corroborated these new findings by analysis of publicly available scRNA-seq data of the adult mouse brain (Tabula Muris; Suppl. Figure 4c).

2. In line with the previous point, the authors did not check the involvement of β 2-tancytes which are located in the basal part of the third ventricle and are in contact with blood vessels.

Tanycytes have been involved in the ME permeability (see i.e. Langlet et al, Cell Metab, 2013), they express other semaphorins and mediate the interaction between GnRH axons and the vasculature (Parkash et al, Nat comm, 2015). Is SEMA6A expressed by the β 2-tanycytes?

Reply: As outlined in our response above, we have now also included a marker for β 2-tanycytes (vimentin) in our immunohistochemical analysis and demonstrate that β 2-tanycytes do not express SEMA6A (Page 10 of Results section and Figure 5d).

Are the β 2-tanycytes altered in the SEMA6A KO mice?

Reply: To address this query, we compared the morphology of β 2-tanycytes in the ME of *Sema6a*^{+/+} and *Sema6a*^{-/-} mice, by vimentin immunostainings and found no alterations in their morphology (Results section at page 15 and Suppl. Figure 6b).

Are there differences in the 3rd ventricle formation? (one could think so from the figure Fig. 2 M, N).

Reply: We believe that the 3rd ventricle is not altered and that the apparent differences in the representative images chosen for the first version of the manuscript likely arise from differences in the cutting/processing of the tissue. In fact, we have many other images that do not show differences in third ventricle size. Therefore, to avoid misleading interpretation we have replaced the indicated panels in a new Figure 2e in the revised version of the manuscript.

3. The TEER experiment in Fig 4F would suggest that the expression of SEMA6A receptors is polarized in endothelial cells, is it the case? Which Plexin receptors are participating in the permeability response in the endothelial cells?

Reply: In response to this point, we followed different strategies. First, by analyzing the single cell transcriptomic profile of brain endothelial cells, we showed that one of SEMA6A canonical receptors, *Plxna2*, is expressed by fenestrated endothelial cells, whereas all the other A type plexins are expressed at negligible levels. Moreover, by using the BulkECexplorer, a compendium that we recently generated of 240 publicly available bulk RNA-seq datasets from five human and mouse EC subtypes, including HUVECs and mBECs, we found that *Plxna2* transcripts were on average ~10-fold higher than *Plxna1*, *Plxna3* or *Plxna4* in both HUVECs and mBECs, with HUVECs showing the highest levels (Suppl. Figure 5f). These results (included at pages 12-13 of Results section) were then further validated by quantitative RT-qPCR in HUVECs (Suppl. Figure 5g). Also, immunofluorescent staining of mBECs combined with high resolution confocal imaging and 3D rendering showed that Plexin-A2 protein did not colocalize with the luminal/apical membrane marker ICAM-2, but that its distribution is rather polarized towards the basal side (Page 13 and Figure 6e).

Can the silencing of the receptors rescue the increase of permeability?

Reply: To address this point, we have performed permeability assays in HUVECs infected with lentiviral particles carrying *PLXNA2* or scrambled control shRNA sequences. We have found that *PLXNA2* knockdown significantly attenuates SEMA6A-induced permeability of HUVECs (Pages 13-14 of Results section and Figure 6f), hence functionally demonstrating that Plexin-A2 participate in SEMA6A-induced vascular permeability.

Is there expression of Plx2A and Plx4A in the ME?

Reply: To address this point, we first analyzed publicly available scRNAseq datasets of adult brain endothelial cells (Tabula Muris and EC atlas) and found that *Plxna2* was enriched in fenestrated ECs (fECs) co-expressing PLVAP (Page 13 of Results section, Figure 6c,d and Suppl. Figure 5e). Further, co-localisation experiments on wild-type adult ME revealed that Plexin-A2 is expressed by PECAM1⁺ blood vessels in the ME (Page 14 of Results section and Figure 7a). Informed by scRNAseq data that did not detect *Plxna4* transcripts in fECs and showed negligible *Plxna4* levels in ECs in general, we did not test its protein expression *in situ* on ME sections.

Also, Fig Suppl 2C and G need a positive control to be sure that the antibody used in the staining worked but the protein is absent in the region.

Reply: Although the antibodies are commercially available and have been previously used by us and others on mouse tissues (ref #77, Plein, A. et al. J. Clin. Invest. 2015; ref #78, Poltavski, D. M. et al. Elife. 2019), we agree that the lack of signal in the images displayed could be misleading. However, we would like to point the reviewer's attention to Plexin-A2 signal that is visible in the image included around the lateral ventricles. Still, to confirm the specificity of the antibodies, we have included additional single immunofluorescence staining at E14.5 showing the presence of specific Plexin-A2 and Plexin-A4 signals in the cortex (Cx), trigeminal ganglion (TG) and pons that nicely match *in situ* hybridisation signals of publicly available gene expression datasets (GenePaint). Of note Plexin-A2 and Plexin-A4 expression in the TG has also been reported by others in Schwarz et al., 2008 (<https://doi.org/10.1016/j.ydbio.2008.08.020>) and Cheng et al., 2001 ([https://doi.org/10.1016/S0896-6273\(01\)00478-0](https://doi.org/10.1016/S0896-6273(01)00478-0)). In addition, we also performed peroxidase-based immunostaining following avidin-biotin amplification that revealed the presence of specific Plexin-A2 signal next to the lateral ventricle and of Plexin-A4 in the corpus callosum (CC)/anterior commissure (AC), thus confirming the specificity of these antibodies. Still, despite amplification steps, Plexin-A2 and Plexin-A4 signals were absent in the MPOA, further confirming lack of Plexin-A2 and Plexin-A4 expression in the MPOA of the brain where GnRH neurons project. We have included these new data in this rebuttal but could upon request also include these results in the Supplementary data.

4. In relation of the point above, it would be important to show raw TEER values instead of relative TEER values versus T0 to better evaluate the tightness of the barrier. I would also suggest to show also a longer time period of TEER measurement to observe the maintenance of the effect overtime.

Reply: To address this point, we have provided the absolute TEER values at T0 for each TEER experiment in the corresponding Figure legends. We have further included additional time points for the TEER experiments, now included in Fig. 6b (Page 12 of Results section) and Suppl. Figure 5d.

5. The authors use the expression of PLVAP as a proxy for vascular permeability (Fig 4), however this result needs a more convincing method to evaluate the leakage in the fenestrated vessels of the ME. The authors should use tracers of different size (such as cadaverine, dextran 10kDa or fluorescent albumin) to assess the changes in vascular permeability.

Reply: We thank the reviewer for the interesting suggestion to use dyes to assess ME permeability *in vivo*. To complement our findings, we have therefore performed Evans Blue dye injections in *Sema6a*^{+/+} and *Sema6a*^{-/-} mice, as done previously by others, to fluorescently label albumin and probe access of peripheral signals through the ME barrier *in vivo* (ref #56, Langlet et al., 2013 and ref #57, Jiang et al., 2020). This analysis revealed a significant reduction in dye diffusion in the ME of *Sema6a*^{-/-} as compared to *Sema6a*^{+/+} mice (Page 14 of Results section and Figure 7b,c). These data strongly support a role for SEMA6A in the control of ME permeability.

6. It would be also important to better characterize the vascularization defects in the ME in the SEMA6AKO at embryonic and adult time points in correlation with other defects observed in the mutant mice. For instance, are there only less loops or also less vessels? I would suggest to use other vascular maskers, such as Glut1 or CD31, since they are more specific than IB4.

Reply: To address this point, we have co-immunostained the ME in adult *Sema6a*^{+/+} and *Sema6a*^{-/-} mice for the endothelial marker PECAM1 (CD31) and the fenestrated vessel marker PLVAP. Our results show not only reduced PLVAP immunostaining but also reduced PECAM1⁺ capillary loops in *Sema6a*^{-/-} ME as compared to controls. This indicates that reduced ME barrier permeability as a result of SEMA6A loss is accompanied by defective structural remodeling of ME capillaries (Pages 14-15 of Results section and Suppl. Figure 6a).

7. The mutation identified in human samples was found mostly in males. Do SEMA6A KO mouse males expose any sexual maturation defect?

Reply: To assess sexual maturation defects in *Sema6a*^{-/-} males, we compared balanopreputial separation (BPS) of *Sema6a*^{+/+} and *Sema6a*^{-/-} mice. We found a significant delay in BPS onset in *Sema6a*^{-/-} mice as compared to controls (Figure 4c). Further, *Sema6a*^{-/-} males displayed reduced GnRH innervation of the ME, a significant decrease in gonadal size, reduced *Lhb* levels in the pituitary, and decreased area of Leydig cells expressing CYP17A1. These new data are presented at pages 8-9 of Results and shown in Figure 4d,f,j-m. We also assessed general fertility of the *Sema6a* population by recording breeding over 9 months (Page 9 of Results section and Suppl. Table 2).

Reviewer #2:

In this study, Lettieri et al. explore the roles of the member of the semaphorin family, SEMA6A, in the control of ME vascular permeability and thereby GnRH function, as putative component for pubertal control. The study is nicely presented and contains a substantial amount of experimental data, including preclinical and clinical findings, which collectively point to a previously undocumented role of SEMA6A in the control of GnRH system, by modulating GnRH innervation of the ME. While the experimental data are convincing and the results are coherently discussed, there are several points that require further attention by the authors.

We thank the Reviewer for this positive assessment.

1. Admittedly, the role of SEMA6A in the control of GnRH neuronal system has not been

previously reported. Yet, other members of the SEMA family, such as SEMA3A and SEMA7A, had been implicated previously in the control of different aspects of GnRH neurons. While the original papers referring to this regulatory role are quoted in the paper, discussion of similarities and differences between SEMA6A and these other members is relatively superficial and deserves more attention. Overall, it is crucial that the authors make a stronger case for the novelty and specificity of the mechanism proposed, as the reader might be left with the impression that SEMA6A is just another element of the complex regulatory pathway of SEMA factors involved in GnRH control, as previously documented.

Reply: To address this point, we have further emphasized the novelty of our findings, including discussing new results that support this novelty. For example, data showing that SEMA6A expressed by pre-myelinating oligodendrocytes regulates vascular permeability at the ME independently of tanycytes. We now discuss more in detail these and other novel aspects of our findings at pages 19-20.

2. The mechanism proposed is related with changes in vascular ME permeability, which may impair the inflow and outflow of multiple factors across the blood brain barrier at the level of the ME. If this is the case, are the authors sure the reported effects are specific of GnRH neurons, or might this be just one of the manifestations of a wider perturbation, as one might expect from such an alteration in the vascular permeability at the ME. The narrow focus on puberty and GnRH neurons might obscure identification of additional targets and mechanisms for the observed phenotype, or might prevent identification of other phenotypes related with impairment of SEMA6A. This needs to be discussed in deeper detail and specificity of the findings, ideally, further supported with additional data.

Reply: To address this point, we now discuss in more detail this matter of specificity in the revised manuscript (page 15 of Results and page 21 of Discussion). We cannot exclude that other neuroendocrine neurons might also be affected by the loss of SEMA6A and the reduction in vascular permeability. Still, our patients did not display additional pituitary hormone deficiencies (which we now explicitly state in the manuscript). One of the criteria used to select our patient cohort was the presence of normal values in Thyroid Function Tests (TFTs) and IGF-1. This strongly suggests that loss of SEMA6A or the patient-specific mutation do not affect all neuroendocrine cells projecting to the median eminence, such as Thyrotropin-Releasing Hormone- and Growth-hormone- releasing neurons, respectively. While it is possible that other neuronal populations are affected, these data confirm at least a level of specificity. In future studies it will be interesting to investigate this further.

3. While considerable focus is placed on the role of SEMA6A on pubertal maturation, the indices used for monitoring puberty in the mouse model are rather limited, even if findings are very convincing. For instance, there is a clear delay in vaginal opening, but this phenomenon is not confronted with analyses of other, more direct parameters of pubertal activation, such as ovarian maturation, first estrus and/or changes in LH and/or estradiol levels. In the absence of such additional parameters, the argumentation of pubertal changes is weakened and the mechanisms underlying the observed pubertal delay remain superficially characterized. This by no means implies the findings are not convincing, but may fall short in proving the major conclusion, as vaginal opening might not sufficiently capture the complexity of pubertal maturation.

Reply: To address this point, we now provide additional data on pubertal activation in *Sema6a*^{-/-} female mice, including, as suggested, first estrus measurements (Figure 4b), ovary size (Figure 4g,h) and ovarian maturation parameters (Figure 4i). Further,

reproductive behaviour has now also been analysed in *Sema6a*^{-/-} male mice. This showed delay in balanopreputial separation (Figure 4c), reduced gonadal size, reduced pituitary *Lhb* levels and reduced area of Leydig cells expressing CYP17A1 (Figure 4j-m). Finally, we have also included results from breeding records of *Sema6a* mutant mice in comparison to those of WT *C57Bl/6J* mice, for a period of 9 months of breeding (Suppl. Table 2). This analysis confirmed the pubertal maturation and reproductive phenotypes of *Sema6a* mutants described above. The new data are presented at pages 8-9 of Results section.

Reviewer #3:

In this study, Lettieri A et al. report a novel loss-of-function mutation in SEMA6A. SEMA6A can be a potential candidate gene for delayed puberty. This finding was confirmed in a validation cohort and supported by functional in vitro and in vivo studies. The study addresses an interesting topic on the pathogenesis of hypogonadism or delayed puberty. Despite these strengths, there are some issues that need careful consideration and eventual revision, as they may weaken the conclusions and general impact of the study.

We thank the Reviewer for these positive comments.

1. The most significant concern of these findings is that the missense variant of the SEMA6A gene is not constrained. The heterozygous variant in this study is unlikely to have a clinical impact.

Reply: We thank the reviewer for this comment, but we respectfully disagree. The GERP score is very high (Figure 8c), with high REVEL and CADD scores (Figure 8b), suggesting that this portion of the gene is highly constrained.

2. In the animal functional study, heterozygous mutant mice showed no difference from wild-type mice in the vaginal opening. It would be good to add more explanation about this in the discussion section.

Reply: To address this point, we have included an additional reference in the section of the Discussion (page 21-22) that discusses this point, as mentioned in the answer above. Thus, previous works on several related genes has demonstrated that even when patients carry heterozygous pathogenic mutants, only full knock-out, but not heterozygous, mice targeting the corresponding genes exhibit a pathological reproductive phenotype (e.g. *SEMA3A*, *SEMA7A*, *PLXNA1*) (see references in the Discussion page 21). Moreover, our breeding records (now included in the Results section at page 9 and Suppl. Table 2) also show defects in heterozygous *Sema6a* suggesting that these mice express reproductive phenotypes albeit not at the same extent.

3. Did the authors confirm this novel variant by sanger sequencing?

Reply: The variant has been confirmed by Sanger sequencing and we have included these data in the revised manuscript (page 15 in Results section and Suppl. Figure 7).

4. Clinical information on patients is lacking. Are all patients "delayed puberty"? Clinical information such as the age of onset of puberty and hormone levels for the patient should be added to the table.

Reply: To address this point, we now provide this additional information in Table 1 of the revised manuscript.

5. The use of COS-7 cells addresses this, but this cell line might not reflect well the GnRH neuronal physiology. I would suggest to confirm using GT1-7 cells.

Reply: Our new results identify oligodendrocytes and not GnRH neurons as the cellular source of SEMA6A (page 10 in Results section and Figure 5e). We therefore feel that testing SEMA6A in a GnRH neuron cell line will not further strengthen our results.

REVIEWERS' COMMENTS

Reviewer #1 (Remarks to the Author):

The revised manuscript "SEMA6A drives GnRH neuron-dependent puberty onset by tuning median eminence vascular permeability" by Lettieri et al. has notoriously improved, mainly by providing a more robust mechanism for semaphorin6a (Sema6a)-mediated regulation of puberty onset. The authors have now identified that the cells expressing Sema6a in the median eminence (ME) are pre-myelinating oligodendrocytes. They also found that the Sema6a receptor expressed by endothelial cells is PlexinA2, which modulates barrier permeability in the ME. Also, the blood-brain barrier experiments, both in vivo and in vitro, are more convincing and strongly support the proposed mechanism. In addition, the characterization of the puberty onset in Sema6a^{-/-} females has been better characterised and the authors have extended the analysis to males, which provides a broader window of the effects of Sema6a in sexual maturation.

The manuscript is well written and carefully executed and, the results are very interesting and novel. Therefore, I recommend its publication.

Reviewer #2 (Remarks to the Author):

The authors have extensively revised their work, following the indications of the referees. Overall, the robustness of the findings has been increased and the mechanistic insight improved. The authors are commended for their efforts to meet the requirements of the referees.

Reviewer #3 (Remarks to the Author):

Many of the points mentioned have been revised to improve the manuscript.

There are no concerns to publish this manuscript.

Point-to-point rebuttal - SEMA6A drives GnRH neuron-dependent puberty onset by tuning median eminence vascular permeability

Reviewer #1 (Remarks to the Author):

The revised manuscript “SEMA6A drives GnRH neuron-dependent puberty onset by tuning median eminence vascular permeability” by Lettieri et al. has notoriously improved, mainly by providing a more robust mechanism for semaphorin6a (Sema6a)-mediated regulation of puberty onset. The authors have now identified that the cells expressing Sema6a in the median eminence (ME) are pre-myelinating oligodendrocytes. They also found that the Sema6a receptor expressed by endothelial cells is PlexinA2, which modulates barrier permeability in the ME. Also, the blood-brain barrier experiments, both in vivo and in vitro, are more convincing and strongly support the proposed mechanism. In addition, the characterization of the puberty onset in Sema6a^{-/-} females has been better characterised and the authors have extended the analysis to males, which provides a broader window of the effects of Sema6a in sexual maturation. The manuscript is well written and carefully executed and, the results are very interesting and novel. Therefore, I recommend its publication.

Reviewer #2 (Remarks to the Author):

The authors have extensively revised their work, following the indications of the referees. Overall, the robustness of the findings has been increased and the mechanistic insight improved. The authors are commended for their efforts to meet the requirements of the referees.

Reviewer #3 (Remarks to the Author):

Many of the points mentioned have been revised to improve the manuscript. There are no concerns to publish this manuscript.

Reply: We would like to thank the reviewers for their positive assessment of our revised manuscript and for acknowledging the efforts made to improve our initial submission with detailed expression studies, more extensive description of the pubertal phenotype and mechanistic insights.